# Vagal determinants of exercise capacity

Asif Machhada[1,2], Stefan Trapp[1], Nephtali Marina[1], Robert C.M. Stephens[3], John Whittle[4], Mark F. Lythgoe[2], Sergey Kasparov[5], Gareth L. Ackland[1,6] & Alexander V. Gourine[1]

Indirect measures of cardiac vagal activity are strongly associated with exercise capacity, yet a causal relationship has not been established. Here we show that in rats, genetic silencing of the largest population of brainstem vagal preganglionic neurons residing in the brainstem's dorsal vagal motor nucleus dramatically impairs exercise capacity, while optogenetic recruitment of the same neuronal population enhances cardiac contractility and prolongs exercise endurance. These data provide direct experimental evidence that parasympathetic vagal drive generated by a defined CNS circuit determines the ability to exercise. Decreased activity and/or gradual loss of the identified neuronal cell group provides a neurophysiological basis for the progressive decline of exercise capacity with aging and in diverse disease states.

[1] Centre for Cardiovascular and Metabolic Neuroscience, Neuroscience, Physiology and Pharmacology, University College London, London WC1E 6BT, UK. [2] UCL Centre for Advanced Biomedical Imaging, Division of Medicine, University College London, London WC1E 6DD, UK. [3] University College London Hospitals NIHR Biomedical Research Centre, London WC1E 6BT, UK. [4] Division of Medicine, University College London, London WC1E 6BT, UK. [5] Department of Physiology and Pharmacology, University of Bristol, Bristol BS8 1TD, UK. [6] Translational Medicine and Therapeutics, William Harvey Research Institute, Queen Mary University of London, London EC1M 6BQ, UK. Correspondence and requests for materials should be addressed to G.L.A. (email: g.ackland@qmul.ac.uk) or to A.V.G. (email: a.gourine@ucl.ac.uk).

Elite endurance athletes display exceptionally high parasympathetic vagal tone[1,2]. Although indirect measures of high cardiac parasympathetic activity correlate with enhanced exercise capacity[3,4] and lower all-cause mortality[5] in athletes and the general population, it remains unclear (and currently contentiously debated[6,7]) whether vagal tone can be enhanced by exercise training. Alternatively, intrinsically high parasympathetic activity, for example due to genetic factor(s), could predispose some individuals to higher tolerance for enduring training regimes, essential to achieve superior athletic performance.

This study was designed to directly test the hypothesis that the strength of vagal tone determines exercise capacity implying that vagal withdrawal should reduce while vagal recruitment should enhance the ability to exercise. We tested it in rodents by genetically targeting the key population of vagal preganglionic neurons that reside in the brainstem's dorsal vagal motor nucleus (DVMN). Developmental studies suggested that the DVMN is the primary vagal nucleus and probably the most (evolutionary) ancient central nervous system (CNS) structure which harbours autonomic neurons in vertebrates[8,9]. Cardiac vagal preganglionic neurons which reside in the DVMN innervate ventricular myocardium[10], but have limited control over the chronotropic function[9]. These neurons are tonically active[11] and critically important for metabolic control[12]. A subset of DVMN neurons provides functional innervation and contributes to the autonomic control of the electrical and contractile properties of the left ventricle (LV) of the heart[13,14].

In this study, we show that genetic silencing of the DVMN neurons dramatically impairs exercise capacity, while optogenetic recruitment of this neuronal population enhances cardiac contractility and prolongs exercise endurance in rats. These data provide direct experimental evidence that parasympathetic drive generated by the DVMN neurons determines the ability to exercise.

## Results

**DVMN neuronal activity and the level of voluntary exercise.** First, electrophysiological recordings of DVMN neuronal activity

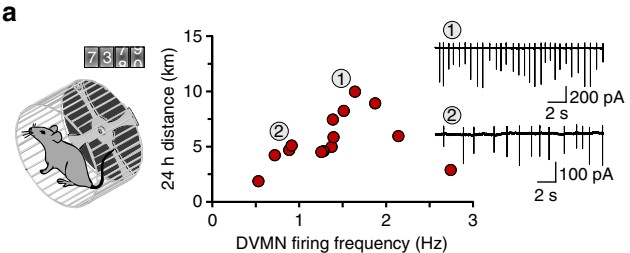

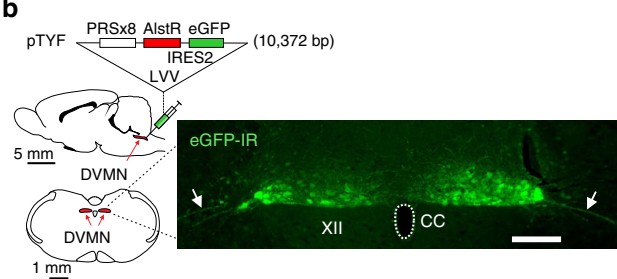

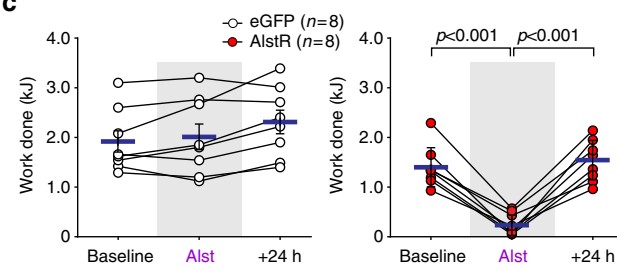

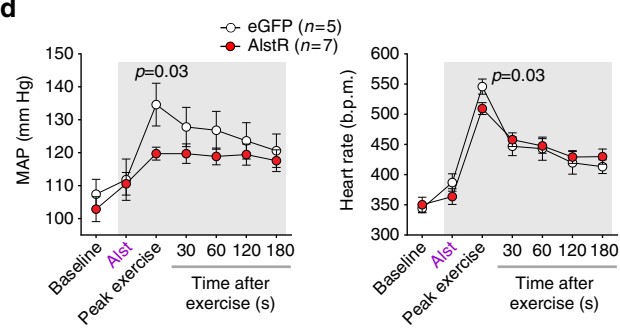

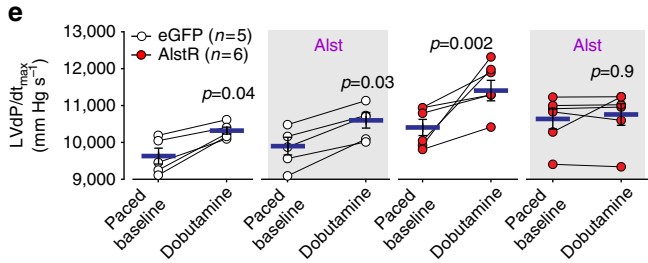

**Figure 1 | Activity of DVMN vagal preganglionic neurons determines exercise capacity.** (**a**) Left: mice were housed in standard home cages with voluntary access to running wheels. Right: association between the intrinsic activity of the DVMN neurons (recorded in brainstem slices in vitro) and the amount of voluntary exercise exhibited by the animals in a 24 h period. Inset: representative examples of the electrical activity of DVMN neurons recorded in cell-attached configuration in brainstem slices of mice which were running 10 km (1) and 5.1 km (2) in a 24 h period. (**b**) Genetic targeting of the DVMN vagal preganglionic neurons in rats. Left: PRSx8-AlstR-IRES2-eGFP-LVV vector layout and schematic drawings of the rat brain in saggital and coronal projections illustrating the anatomical position of the DVMN, Right: photomicrograph of a representative coronal section of the dorsal brainstem illustrating the distribution of transduced DVMN neurons (eGFP amplified by immunohistochemistry) in the caudal region of the nucleus (Bregma level: −13.8 mm). Arrows point at ventrally projecting axons of the transduced DVMN neurons (forming the efferent vagus nerve). XII, hypoglossal motor nucleus (cells are not visible); eGFP-IR, eGFP immunoreactivity; CC, central canal. Scale bar: 200 μm. (**c**) Summary data illustrating the effect of allatostatin (Alst) infusion into the cisterna magna on exercise capacity in rats transduced to express eGFP or AlstR/eGFP by the DVMN neurons. Data are presented as individual values and means ± s.e.m. Comparisons are made using ANOVA. (**d**) The effect of allatostatin on exercise-induced changes in systemic arterial blood pressure and heart rate in rats transduced to express eGFP or AlstR/eGFP by the DVMN neurons. MAP, mean arterial blood pressure. Data are presented as means ± s.e.m. Comparisons are made using ANOVA. (**e**) Summary data illustrating cardiac contractile responses (determined by changes in the first differential of left ventricular pressure, LVdP/dt$_{max}$) to β-adrenoceptor stimulation (administration of dobutamine) at baseline (paced at 20% above the resting heart rate) and after allatostatin infusion into the cisterna magna in rats transduced to express eGFP or AlstR/eGFP by the DVMN neurons. Data are presented as individual values and means ± s.e.m. Comparisons are made using paired t-test.

were made in brainstem slices of adult mice that were housed in standard home cages with voluntary access to the running wheels (Fig. 1a). One hundred fifteen DVMN neurons recorded *in vitro* (that is, in the absence of afferent modulation from the periphery and the rest of the CNS) from 14 mice exhibited a mean spontaneous firing rate of $1.46 \pm 0.12$ Hz. The mean DVMN neuronal action potential firing rate (determined for each mouse; 8 cells on average) in the range $0–1.7$ Hz increased with the amount of voluntary exercise performed by the animals in a 24 h period (Fig. 1a), supporting the hypothesis that intrinsic vagal activity may determine the ability to exercise. This largely linear relationship did not continue with spontaneous mean firing rates of DVMN neurons above 2 Hz (recorded in brainstem slices of two animals; Fig. 1a). Although, at present the reasons underlying this biphasic relationship remain unclear, the lower amount of voluntary exercise associated with a higher discharge rate of the DVMN neurons could be potentially explained by negative inotropic[14] and chronotropic influences as well as non-cardiac effects of high vagal activity. Considering that the majority of the vagal

projections originating from the DVMN innervate visceral targets, it is conceivable that higher discharge rate of the DVMN neurons mimics the post-prandial state and, thus, may reduce the motivation to exercise. As voluntary exercise is dependent on motivation, and motivation is determined by a multitude of factors, the design of the subsequent studies employed forced exercise experimental paradigm.

**The effect of DVMN neuronal inhibition on exercise capacity.** Next, in rats, highly specific inhibition or recruitment of vagal activity was achieved by targeting the DVMN neurons using viral vectors to express either an inhibitory $G_i$-protein-coupled *Drosophila* allatostatin receptor[15] (AlstR; Fig. 1b) or light-sensitive optogenetic actuator ChIEF[16] (Fig. 2a). Insect peptide allatostatin rapidly inhibits autonomic neurons transduced to express AlstR[17,18] while 445 nm light pulses trigger precisely timed depolarizations and action potential firing of DVMN neurons expressing ChIEF[18]. Exercise capacity (expressed as work done in Joules, J) was determined in an experimental paradigm employing forced exercise on a single lane rodent treadmill following selection for compliance and three 24-h spaced acclimatizing training sessions. After the initial training period, rats expressing AlstR and control transgene (enhanced green fluorescent protein, eGFP) achieved similar baseline exercise capacities (Fig. 1c). Inhibition of the DVMN neurons expressing AlstR following infusion of allatostatin into the *cisterna magna* dramatically reduced the exercise capacity (from $1.40 \pm 0.15$ to $0.26 \pm 0.08$ kJ; $P < 0.001$; ANOVA; Fig. 1c). Exercise capacity fully recovered within the next 24 h ($1.52 \pm 0.15$ versus $1.40 \pm 0.15$ kJ at baseline; $P = 0.7$; ANOVA; Fig. 1c). Peak increases in the arterial blood pressure (ABP) and heart rate during exercise were blunted in conditions of DVMN silencing ($120 \pm 2$ versus $135 \pm 7$ mm Hg in controls; $P = 0.03$; ANOVA and $510 \pm 7$ versus $546 \pm 12$ b.p.m. in controls; $P = 0.03$; ANOVA, respectively; Fig. 1d). These data

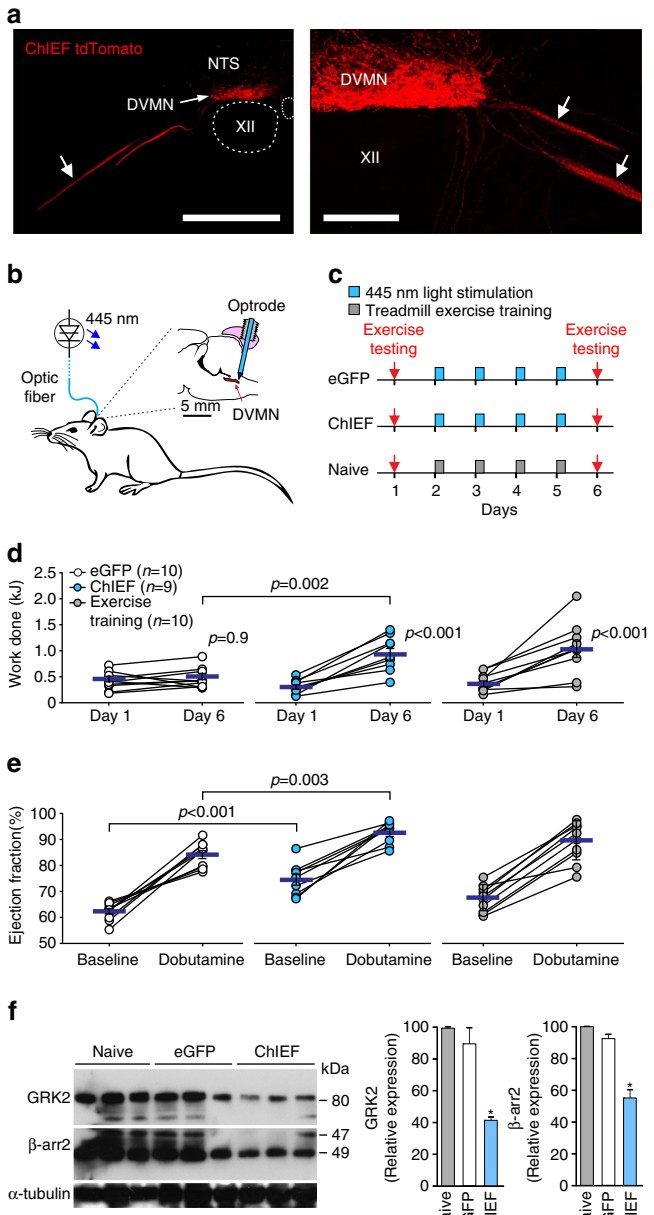

**Figure 2 | Recruitment of vagal activity increases exercise capacity.**
(**a**) Photomicrographs of coronal sections of the rat brainstem taken at low (*left*) and high (*right*) magnification illustrating representative examples of CHIEFtdTomato expression in the caudal region of the DVMN (Bregma level: $-13.8$ mm) 6 weeks after microinjections of PRSx8-ChIEFtdTomato-LVV. Neurons display specific membrane localization of the transgene. Arrows point at ventrally projecting axons of the transduced DVMN neurons (forming the efferent vagus nerve). XII, hypoglossal motor nucleus; NTS, nucleus of the solitary tract. Scale bars: 500 μm (*left*), 200 μm (*right*). (**b**) Schematic drawing of the experimental setup. The rat brain drawn in saggital projection shows the approximate positioning of the chronic optrode implant. (**c**) Illustration of the experimental protocols. ChIEF, animals transduced to express ChIEFtdTomato in the DVMN. eGFP, animals transduced to express eGFP in the DVMN. (**d**) Exercise capacity of rats transduced to express eGFP or ChIEFtdTomato by the DVMN neurons before and after 4 days of light stimulation of the dorsal brainstem as well as of a separate group of naive animals before and after 4 daily sessions of treadmill exercise training. Data are presented as individual values and means ± s.e.m. Comparisons are made using ANOVA. (**e**) Summary data showing LV ejection fraction at baseline and in response to β-adrenoceptor stimulation (administration of dobutamine) in rats transduced to express eGFP or ChIEFtdTomato by the DVMN neurons after 4 days of light stimulation of the dorsal brainstem and in naive animals following four daily sessions of exercise training. Data are presented as individual values and means ± s.e.m. Comparisons are made using ANOVA. (**f**) *Left*: representative immunoblots showing GRK2 and β-arrestin 2 expression in LV lysates of rats transduced to express eGFP or ChiEFtdTomato by the DVMN neurons after 4 days of light stimulation of the dorsal brainstem. *Right*: summary data illustrating means ± s.e.m. of the densitometry of GRK2 and β-arrestin 2 relative expression. $*P < 0.05$ (ANOVA).

suggested that vagal drive generated by the DVMN neurons is not simply associated with, but determines the ability to exercise.

We next determined whether reduced activity of the DVMN neurons may alter cardiac contractile responses to sympathetic β-adrenoceptor-mediated stimulation, which is essential to trigger and maintain appropriate increases in cardiac output to support circulatory requirements of exercise. In anaesthetized (urethane 1.3 g kg$^{-1}$) rats expressing either eGFP or AlstR in the DVMN neurons, systemic administration of the β$_1$-adrenoceptor agonist dobutamine (5 µg kg$^{-1}$, intravenous (i.v.)) increased the maximum differential of LV pressure (LVd$P$/d$t_{max}$; Fig. 1e) and LV end systolic pressure (LVESP; Supplementary Fig. 1), indicative of increased inotropy. Dobutamine had a similar effect on LVd$P$/d$t_{max}$ (increase by 700 mm Hg s$^{-1}$; $P = 0.03$; paired $t$-test; Fig. 1e) and LVESP (increase by 13 mm Hg; $P = 0.006$; paired $t$-test; Supplementary Fig. 1) in rats expressing eGFP in the DVMN following allatostatin infusion. However, when DVMN neurons were silenced, β-adrenoceptor stimulation had no effect on LVd$P$/d$t_{max}$ (increase by 122 mm Hg s$^{-1}$; $P > 0.9$; paired $t$-test; Fig. 1e) and LVESP (increase by 1 mm Hg; $P > 0.9$; paired $t$-test; Supplementary Fig. 1), suggesting that reduced vagal drive from the DVMN impairs the ability of the heart to mount a contractile response to sympathetic stimulation.

**The effect of DVMN neuronal stimulation on exercise capacity.** In a reverse experimental paradigm, we determined whether selective and highly specific optogenetic recruitment of the DVMN activity (Fig. 2b) could increase exercise capacity. Rats were selected on the basis of relatively poor baseline exercise endurance and randomized into three experimental groups. Exercise capacity was determined twice: on day 1 (baseline) and on day 6 to determine the effect of treatment. During the 4 day interim period (see timeline Fig. 2c), the dorsal brainstem of rats expressing optogenetic actuator ChIEF ($n = 9$) or control transgene eGFP ($n = 10$) in the DVMN neurons was illuminated via an implanted optrode with 445 nm light (10 ms, 15 Hz) for 15 min/session (Fig. 2b). The third group of naive (non-transduced) animals ($n = 10$) was subjected to daily sessions of treadmill exercise training (~15 min long). Four daily sessions of vagal stimulation by optogenetic recruitment of the DVMN activity was sufficient to double the exercise capacity (0.94 ± 0.11 versus 0.47 ± 0.06 kJ in rats expressing eGFP; $P = 0.002$; ANOVA; Fig. 2d, middle). This improvement in exercise capacity was comparable to that observed in the naive rats trained to exhaustion over the same time period (Fig. 2d, right). Ultrasound assessment of LV function demonstrated that optogenetic DVMN stimulation increased baseline ejection fraction (75 ± 2% versus 62 ± 1% in rats expressing eGFP; $P < 0.001$; ANOVA; Fig. 2e) and enhanced contractile response to β-adrenoceptor stimulation

with dobutamine (93 ± 1 versus 84 ± 2% in rats expressing eGFP; $P = 0.003$; ANOVA; Fig. 2e). Resting heart rate and chronotropic responses to dobutamine were similar between the experimental groups (Supplementary Fig. 2).

Increased cardiac contractility (at rest and in response to dobutamine challenge) following optogenetic stimulation of the DVMN neurons suggested that the heart became more responsive to sympathetic β-adrenoceptor stimulation. Western blot analysis revealed that four daily sessions of vagal stimulation by optogenetic recruitment of the DVMN activity resulted in a significant downregulation of left ventricular G-protein-coupled receptor kinase 2 (GRK2) and β-arrestin 2 expression (Fig. 2f). GRKs promote phosphorylation of the intracellular domains of β-adrenoceptors, recruiting arrestins to block receptor couplling to G-proteins resulting in receptor desensitization and internalization[19]. These data suggest that the activity of the DVMN neuronal projections has a major impact on LV inotropic state by controlling expression of key negative regulators of β-adrenoceptor-mediated signalling.

**Cardiac vagal activity and aerobic capacity in humans.** The data obtained in rodents suggested that vagal drive powerfully modulates cardiac reactivity to catecholamines and determines exercise capacity. To investigate whether these conclusions are consistent with the determinants of human exercise performance, we next examined whether cardiac vagal activity correlates with aerobic capacity in humans. Although the strength of parasympathetic tone to the LV in humans is impossible to measure directly, the rate of heart rate recovery (HRR) after cessation of the exhaustive exercise serves as a robust index of individual ability to recruit vagal tone (Fig. 3a) and is highly sensitive to muscarinic blockade[20]. Assessment of HRR avoids deriving parasympathetic activity from measures dependent on absolute heart rate—a major confounder in heart rate variability analysis[21]. We determined HRR in 1293 human participants within an age range associated with the onset of vagal dysfunction (63 ± 14 years of age; BMI: 26.9 ± 5.3; 67.5% males). We found a graded relationship between HRR and independent measures of aerobic performance including peak oxygen delivery (Fig. 3b), anaerobic threshold (A$_T$; Fig. 3c), predicted work rate (Fig. 3d), $V_E/V_{CO_2}$ (inversely related to cardiac output at peak exercise[22]; Supplementary Fig. 3) and oxygen pulse (Supplementary Fig. 3),—a robust surrogate for LV stroke volume.

**Discussion**
Exercise capacity ultimately determines individual ability to chase prey, escape from predators, explore new environments or simply enjoy physical activity. The human exercise performance data and the results of experimental animal studies reported here provide

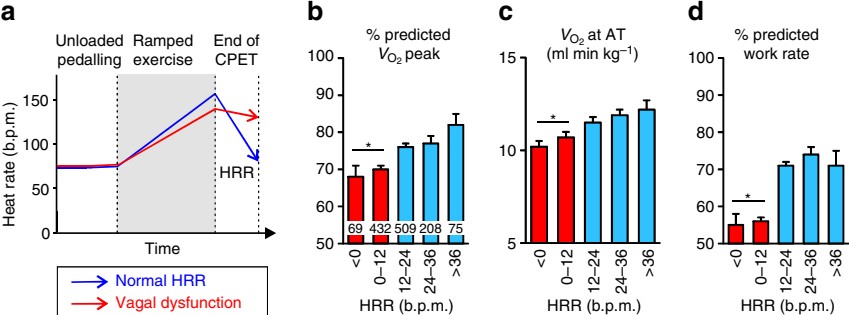

**Figure 3 | Exercise capacity is reduced in subjects with cardiac parasympathetic dysfunction.** (**a**) Schematic of the CPET protocol in humans. Percentage of predicted peak oxygen consumption (**b**), A$_T$ (**c**) and percentage of predicted work rate (**d**) in subjects ($n = 1,293$), stratified by the speed of HRR 1 min after cessation of CPET. Numbers of subjects are shown within bars. Data are presented as means ± s.e.m. *$P < 0.001$ (ANOVA).

the first direct evidence that the strength of the parasympathetic vagal drive determines cardiac contractility and exercise capacity. Differences in individual ability to exercise may, therefore, reflect varying activity levels of the DVMN neuronal projections. We hypothesize that high parasympathetic activity generated by the DVMN neurons maintains the ability of the heart to mount an augmented contractile response to sympathetic stimulation and increases the 'operational range' of the heart by downregulating GRK2 and arrestin expression in ventricular myocytes. This tonic vagal influence originating from the DVMN appears to be independent of relatively modest direct acetylcholine-mediated negative inotropic effect[14] and chronotropic control of the heart, which is provided by another notable group of vagal preganglionic neurons residing in the nucleus ambiguus[9]. Vagal preganglionic neurons that innervate the LV are located in the caudal region of the left DVMN[14]. Since DVMN neurons provide parasympathetic innervation of the respiratory system and various visceral organs, in order to preferentially recruit DVMN cardiac projections we targeted the caudal aspects of the nucleus. Although, DVMN inhibition or activation resulted in functional and transcriptional changes at the level of the myocardium, it is plausible that some of the observed changes in cardiac physiology might be due to the inhibition/recruitment of vagal efferent projections to other targets[23] and recruitment of circulating cardiotropic factor(s)[24].

Higher parasympathetic vagal tone in elite athletes is suggested to confer higher tolerance for intense exercise regimes and might be further enhanced by unconventional stimuli, including peripheral sensory (nociceptive) stimulation. For example, cycle(s) of ischaemia/reperfusion applied to the limb appear to recruit DVMN activity (as shown in an animal study[18]), increase cardiac vagal activity in healthy volunteers[25], and improve exercise performance in highly trained athletes[26]. An ability to respond to noxious stimuli (usually associated with threat and/or injury) with increased endurance seems to have a clear evolutionary advantage. On the other hand, DVMN neurons are known to be highly susceptible to metabolic stress[11] and their dysfunction is associated with a host of autonomic abnormalities[11,27,28]. The data obtained in the present study also suggest that loss of DVMN activity may play a critical role in the progressive decline of cardiac contractility and exercise capacity during aging and in disparate disease states.

## Methods

All the experiments were performed in accordance with the European Commission Directive 2010/63/EU (European Convention for the Protection of Vertebrate Animals used for Experimental and Other Scientific Purposes) and the UK Home Office (Scientific Procedures) Act (1986) with project approval from the Institutional Animal Care and Use Committee. The human study was approved by the NRES Committee London (Camden & Islington; MREC: 12/LO/0453).

**Recording the activity of the DVMN neurons in vitro.** Coronal (200 μm) brainstem slices were obtained from 4–6 months old male C57BL/6J mice terminally anaesthetized with halothane. Brains were dissected in ice–cold low [$Na^+$] solution containing 200 mM sucrose, 2.5 mM KCl, 28 mM $NaHCO_3$, 1.25 mM $NaH_2PO_4$, 3 mM pyruvate, 7 mM $MgCl_2$, 0.5 mM $CaCl_2$ and 7 mM glucose (pH 7.4). After recovery at 34 °C for 30 min in a solution containing: 118 mM NaCl, 3 mM KCl, 25 mM $NaHCO_3$, 1.2 mM $NaH_2PO_4$, 7 mM $MgCl_2$, 0.5 mM $CaCl_2$ and 2.5 mM glucose (pH 7.4), slices were kept at 34 °C in artificial cerebrospinal fluid (aCSF) containing 118 mM NaCl, 3 mM KCl, 25 mM $NaHCO_3$, 1 mM $MgCl_2$, 2 mM $CaCl_2$ and 10 mM glucose saturated with 95% $O_2$ and 5% $CO_2$ (pH 7.4).

The recording pipettes (3–6 MΩ) were pulled from thin-walled borosilicate capillaries. Recordings were performed in a cell-attached configuration using an EPC-9 amplifier and Pulse/Pulsefit software (Heka Elektronik, Lambrecht, Germany). Electrodes were filled with 120 mM K-gluconate, 5 mM HEPES, 5 mM BAPTA, 1 mM NaCl, 1 mM $MgCl_2$, 1 mM $CaCl_2$ and 2 mM $K_2ATP$ (pH 7.2). Recordings were conducted in aCSF saturated with 95% $O_2$ and 5% $CO_2$ at 28–32 °C. Currents were filtered at 1 kHz and digitized at 3 kHz. The mean action potential frequency was determined by taking the average frequency over a period

of 5 min. Recordings were analysed using Strathclyde Electrophysiology Software (WinEDR/WinWCP; University of Strathclyde).

**Genetic targeting of the DVMN vagal preganglionic neurons.** Vagal preganglionic neurons of the DVMN express the transcriptional factor Phox2 and were targeted to express AlstR, a chimeric channelrhodopsin variant ChIEF fused with a fluorescent protein tdTomato (ChIEFtdTomato), or eGFP (control) using lentiviral vectors (LVV) under the control of a Phox2-activated promoter PRSx8 (ref. 29). Validation of the vectors specificity in transducing the DVMN neurons has been described in detail previously[18]. Application of allatostatin produces rapid, selective inhibition of the autonomic (including vagal preganglionic) neurons expressing AlstR[17,18]. Pulses of 445 nm light trigger precisely timed depolarizations and action potential firing of the DVMN neurons transduced to express ChIEFtdTomato[18].

Young male Sprague–Dawley rats (50–60 g) were anaesthetized using a combination of ketamine (60 mg kg$^{-1}$; intramuscular (i.m.)) and medetomidine (250 μg kg$^{-1}$; i.m.). Adequate depth of surgical anaesthesia was maintained and confirmed by the absence of a withdrawal response to a paw pinch. With the head held within a stereotaxic frame and fixed in prone position, lidocaine (2% solution) was injected subcutaneously for pre-operative analgesia before a midline dorsal neck incision was made to expose the dorsal surface of the brainstem. DVMN neurons spanning the rostro-caudal extent of the left and right nuclei were targeted with two microinjections on each side (0.25 μl at a rate of 0.05 μl min$^{-1}$) of a solution containing viral particles of PRSx8-AlstR-IRES2-eGFP-LVV or PRSx8-eGFP-LVV. To avoid transfection of the dorsal group of A2 neurons, microinjections were made ~0.1 mm ventral to the DVMN, restricting diffusion of viral particles to the neighbouring NTS whilst sparing hypoglossal motoneurons, which do not express Phox2 (Figs 1b and 2a). Taking the calamus scriptorius as the reference point, the microinjections were made at 0.5 mm rostral, 0.6 mm lateral, 0.8 mm ventral and at 1.0 mm rostral, 0.8 mm lateral, 0.6 mm ventral. For optogenetic vagal stimulation, DVMN neurons were targeted using the same approach but with one microinjection of PRSx8-ChIEFtdTomato-LVV on each brainstem side using the caudal set of coordinates. Using atipamezole (1 mg kg$^{-1}$; i.m.) for reversal of anaesthesia, rats were then placed on a regime of post-operative analgesia involving administration of buprenorphine (0.05 mg kg$^{-1}$ d$^{-1}$; s.c.) for 3 days and carprofen (4 mg kg$^{-1}$ d$^{-1}$; intraperitoneal (i.p.)) for 5 days.

**Cannula/optrode implantations.** Three to four weeks after the microinjections of viral vectors into the DVMN, the animals were anaesthetized using a combination of ketamine (60 mg kg$^{-1}$; i.m.) and medetomidine (250 μg kg$^{-1}$; i.m.) and either a 26-gauge guide injection cannula (Plastic Products; in animals expressing AlstR or eGFP) or conically tipped optrode (Art Photonics; in animals expressing ChIEFtdTomato or eGFP) was stereotaxically implanted to reach the dorsal aspect of the cisterna magna. Two small screws were placed into the skull, and the implant was secured in place with dental acrylic. The guide cannula was closed with a dummy cannula that extended from the tip of the guide cannula by ~0.2 mm. Anaesthesia was reversed with atipamezole (1 mg kg$^{-1}$; i.m.). Carprofen was given for post-operative analgesia and the animals were allowed to recover for at least 7 days before the experiments. Rats were 300–350 g at the time of the main study.

**Biotelemetry transmitter implantation.** Biotelemetry was used to record systemic ABP and heart rate in exercising animals. Rats were anaesthetized using a combination of ketamine (60 mg kg$^{-1}$; i.m.) and medetomidine (250 μg kg$^{-1}$; i.m.) and a laparotomy was performed to expose the abdominal aorta. A catheter connected to a telemetry pressure transducer (model TA11PA-C40, Data Science International) was advanced centrally into the aorta and secured with Vetbond (3M). The transmitter was secured to the abdominal wall and the incision was closed by successive suturing of the abdominal muscle and skin layers. Anaesthesia was reversed with atipamezole (1 mg kg$^{-1}$; i.m.); carprofen was given and the animals were returned to their home cages where they were allowed to recover for at least 7 days before the experiments.

**Exercise model.** The exercise capacity of rats was determined in a forced exercise experimental paradigm using a single lane rodent treadmill (Panlab Harvard Apparatus) with an electrical shock grid set to the minimum perceived threshold (0.1 mA). The animals were selected on the basis of their treadmill compliance and exercise capacity and subjected to daily recruitment/training sessions involving running speeds of 20–30 cm s$^{-1}$ over a 5 min period after 15 min of acclimatization to the treadmill environment. To determine the exercise capacity, treadmill speed was raised from 20 cm s$^{-1}$ in increments of 5 cm s$^{-1}$ every 5 min until the animal's hind limbs made contact with the grid four times within a two minute period—the humanely defined point of exhaustion. The distance covered by the animal was recorded and exercise capacity expressed as work done in Joules.

**Assessment of cardiac contractility.** The animals were anaesthetized with urethane (1.3 g$^{-1}$ kg$^{-1}$; i.p.; following induction with 4% isoflurane). Adequate level of anaesthesia was ensured throughout the experiment by continuous monitoring of heart rate, and the ABP stability as well as the absence of a withdrawal

response to a paw pinch. With an animal in a supine position, the femoral artery and both femoral veins were cannulated for the measurements of the systemic ABP, fluid infusion and administration of drugs. The trachea was cannulated and the animal was allowed to free breathe a supplied gas mixture. Body temperature was maintained with a servo-controlled heating pad at $37.0 \pm 0.5\,°C$. Partial pressures of $O_2$ and $CO_2$ as well as pH of the arterial blood were measured every hour (RAPIDLab 348EX, Siemens). If required, oxygen was added to the respiratory gas mixture to maintain blood oxygenation within the physiological range. A 2F Millar pressure catheter (SPR-320NR, Millar Instruments) was advanced via the right carotid artery and placed within the chamber of the LV allowing recordings of LV pressure (LVP). Averaged waveforms of LVP and systemic ABP were used to derive the LVESP and the mean arterial blood pressure (MAP). The maximum limit of the first differential of LV pressure ($LVdP/dt_{max}$) was derived from the LVP recording as an index of inotropic function. Atrial pacing at 20% above the resting heart rate was performed using a 1.6F octapolar electrophysiology catheter (EPR-802, Millar Instruments) positioned in proximity to the right atrium via the jugular vein. Data were recorded using a Power1401 interface and analysed off-line using *Spike2* software (Cambridge Electronic Design).

In some of the experiments, echocardiographic assessment of LV function was conducted using a Vevo 2100 high-resolution ultrasound system with an MS250 13–24 MHz linear array transducer (VisualSonics). Baseline LV ejection fraction and heart rate (lead II ECG) were determined in rats kept under urethane ($1.3\,g^{-1}\,kg^{-1}$; i.p.; following induction with 4% isoflurane) anaesthesia. Ejection fraction was determined using a B-mode acquisition of a parasternal long-axis view of the LV measuring the length of the ventricle during systole and diastole. Three short-axis images perpendicular to this were acquired along the length of the LV to segment the endocardial border. Simpson's rule was applied to approximate the changes in LV volume[30].

**Histology.** At the end of the experiments, the rats were administered with an overdose of pentobarbitone sodium ($200\,mg\,kg^{-1}$, i.p.) and perfused through the ascending aorta with 0.9% NaCl solution followed by 500 ml of 4% phosphate-buffered (0.1 M, pH 7.4) paraformaldehyde; the brains were removed and stored in the same fixative overnight at $4\,°C$. Following cryoprotection (30% sucrose) of the isolated brainstem, a sequence of transverse slices (30 μm) was cut to determine the extent of AlstR/eGFP, ChiEFtdTomato or eGFP expression by the DVMN neurons.

**Immunoblotting.** LV tissue was excised, snap frozen in liquid nitrogen and stored at $-80\,°C$. GRK2 and β-arrestin 2 were immunodetected in cell lysates (whole cell fractions) using anti-GRK2 antibody (C-9; sc-13143; Santa Cruz; 1:1,000 dilution) and anti-β-arrestin 2 antibody (C16D9, 3857; Cell Signalling Technology; 1:1,000 dilution). Proteins were electrophoretically separated in SDS-PAGE gels and transferred to polyvinylidene difluoride membranes (Amersham Biosciences). After antibody labelling, detection was performed (ECL detection system, Amersham Biosciences). Densitometry was used to calculate the level of expression normalized to the expression of α-tubulin to control protein loading. Uncropped images of the western blots are shown on Supplementary Fig. 4.

**Experiment 1.** To determine whether the level of voluntary exercise correlates with the activity of the DVMN neurons, fourteen adult male C57BL/6J mice (age 4–6 months; JAX™ Stock Number 000664; Charles–River Laboratories) were housed one per cage in a room maintained at a constant temperature of $22 \pm 1\,°C$, with a 12:12 h light–dark cycle (with lights onset at 0600 h). Drinking water and Laboratory Rodent Chow were provided *ad libitum*. After 1 week of acclimatization to the new housing facility (following delivery from the supplier), running wheels (Vet-Tech Solutions) were installed in all the cages. Distances run by individual mice were recorded. After 4–6 weeks of housing with voluntary access to the running wheels, activities of the DVMN neurons (on average 8 per animal) were recorded *in vitro* in acute brainstem slices.

**Experiment 2.** To determine the effect of DVMN silencing on exercise capacity, rats transduced to express AlstR ($n = 8$) or eGFP ($n = 8$) by the DVMN neurons were selected for their exercise compliance and underwent three daily recruitment/ training sessions. Following the initial training period, their exercise capacity was determined over the following three days: on day 1 to determine baseline exercise capacity; on day 2 to determine exercise capacity in conditions of DVMN silencing (15 min following delivery of allatostatin peptide [Ser-Arg-Pro-Tyr-Ser-Phe-Gly-Leu-NH2; 4 μl of 100 μM solution in aCSF] into the *cisterna magna*) or sham-treatment (in rats expressing eGFP by the DVMN neurons); and on day 3 to determine exercise capacity after the recovery from the experimental treatment. Microinjections into the *cisterna magna* were performed via an internal injection cannula connected to a PE-20 tubing attached to a 50 μl syringe (Hamilton) driven by a micro-infusion pump (model Sp210iw, World Precision Instruments). In a separate experiment, changes in the ABP and heart rate during and after the exercise were determined in seven rats expressing AlstR and in five rats expressing eGFP by the DVMN neurons following allatostatin administration. For the assessment of myocardial contractility, rats transduced to express AlstR ($n = 6$) or eGFP ($n = 5$) by the DVMN neurons were anaesthetized with urethane and prepared as described above. After the surgical preparation, the rats were left to

stabilize for 15–30 min. The heart was paced at 20% above the resting heart rate. To determine the effect of DVMN inhibition on cardiac responses to β-adrenoceptor stimulation, dobutamine was given intravenously (infusion $2.5\,μg\,kg^{-1}\,min^{-1}$ i.v.; for 2 min at a rate of $0.1\,ml\,min^{-1}$) before and 15 min after allatostatin administration into the *cisterna magna* (4 μl of 100 μM solution).

**Experiment 3.** To determine the effect of optogenetic DVMN stimulation on exercise capacity, rats were pre-selected on the basis of a relatively poor baseline exercise capacity and randomized in three experimental groups. Exercise capacity was determined at two time points: on day 1 (baseline) and on day 6 to determine the effect of treatment. In the interim 4 days, dorsal brainstem of rats expressing ChiEFtdTomato ($n = 9$) or eGFP ($n = 10$) by the DVMN neurons was illuminated via an implanted optrode using 445 nm light (10 ms, 15 Hz) for 15 min under isoflurane sedation (1–1.5% isoflurane, 1:1 $O_2$/air mixture). A group of naive animals ($n = 10$) underwent 4 sessions of treadmill exercise training over four consecutive days with each session lasting up to the defined point of exhaustion. To determine the effect of enhanced DVMN activity and exercise training on inotropic response induced by β-adrenoceptor stimulation, LV ejection fraction and heart rate were determined before and after dobutamine challenge.

**Experiment 4.** To conduct cardiopulmonary exercise testing (CPET), ethical approval (MREC: 12/LO/0453) was obtained from the UK National Information Governance Board and in accordance with the Declaration of Helsinki. Participants were enrolled prospectively at the University College London Hospitals, UK (December 2007–May 2014), referred by their clinical team for CPET as part of the routine assessment of exercise capacity. Participants gave written consent to use CPET data. Exclusion criteria for CPET were in accordance with the international guidelines[31]. Patients with/without delayed HRR had similar cardiovascular, renal and nutritional profiles (Supplementary Tables 1 and 2).

Blinded investigators carried out all exercise testing. Equipment was calibrated before each test using standard reference gases. Subjects undertook CPET on an electronic cycle ergometer (Zan, nSpire or Lode) to maximal tolerance, adhering to the protocol illustrated in Fig. 3a. Continuous 12-lead ECG was recorded. Continuous breath-by-breath gas exchange analysis was performed. All subjects were instructed to continue cycling until symptom-limited fatigue. CPET data were analysed blinded to HRR. $A_T$ was determined by two independent assessors blinded to autonomic parameters and in accord with published guidelines using the modified V-slope method and confirmed by ventilatory equivalents for carbon dioxide ($V_E/V_{CO_2}$) and oxygen ($V_E/V_{O_2}$; refs 32,33). Peak oxygen consumption ($V_{O_2peak}$), oxygen pulse and $V_E/V_{CO_2}$ were determined. $V_E/V_{CO_2}$ is inversely related to cardiac output at peak exercise[22]. Predicted values, which take into the account age, gender, weight and height, were calculated for $V_{O_2peak}$(ref. 34) and oxygen pulse[35]—a robust measure of LV stroke volume[36,37]. Percent-predicted peak $V_{O_2}$ values (derived from the Wasserman/Hansen equations) provide superior measures of these variables[38]. Vagal dysfunction was defined from HRR at 1 min following the cessation of exercise. We binned HRR values in increments of 5 beats $min^{-1}$. A failure to decrease heart rate by $\geq 12$ beats $min^{-1}$ in one minute after the end of exercise was used as a cut-off value, since published data show a robust association between HRR of $\leq 12$ beats $min^{-1}$ and increased risk of cardiovascular death[39,40].

**Data analysis.** Recordings of the cardiovascular variables were analysed using *Spike2* software (Cambridge Electronic Design). Differences between the experimental groups were assessed using GraphPad Prism 6 software. Comparisons were made using two-way ANOVA (followed by *post hoc* Tukey–Kramer testing) or Student's paired or unpaired *t*-test, as appropriate. For the analysis of data obtained in the human study, the variables were compared according to the HRR bin using general linear models of analysis of variance (quantitative variables), controlling for age as a covariate where CPET variables are not referenced to population norms. Data are reported as individual values and means ± s.e.m. Differences with $P < 0.05$ were considered to be significant.

**Data availability.** The data that support the findings of this study are available from the corresponding authors upon request.

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

## Acknowledgements

This study was supported by the British Heart Foundation (A.V.G., G.L.A.; Ref: RG/14/4/30736), The Wellcome Trust (A.V.G.; Refs 095064 and 200893), Medical Research Council (S.K; Ref: MR/L020661/1), Academy of Medical Sciences/Health Foundation Clinician Scientist Fellowship (G.L.A.), and Royal College of Anaesthetists/British Journal of Anaesthesia Basic Science Career Development Award (G.L.A.). MB PhD funding for A.M. was provided by the Medical Research Council and The Rosetrees Trust. A.V.G. is a Wellcome Trust Senior Research Fellow.

## Author contributions

A.V.G., S.T., G.L.A. designed research; A.M., S.T., N.M., R.C.M.S., J.W., G.L.A. and A.V.G. performed research; S.K. contributed unpublished reagents/analytic tools; A.M., S.T., R.C.M.S., J.W., M.F.L., S.K. and G.L.A. analysed data; A.V.G. and G.L.A wrote the paper; A.M., S.T. and S.K. revised the article critically for important intellectual content.

## Additional information

**Competing interests:** The authors declare no competing financial interests.

