## [Peer Review File · Nature Communications]

Reviewers' comments:

Reviewer #1 (expert in neural regulation of the heart)

Remarks to the Author:

General comments

The stated objective of this study was to evaluate the involvement of cardiac vagal activity in the exercise capacity. The results showed that in experimental models, silencing of brainstem vagal preganglionic neurons (dorsal vagal motor nucleus-DVMN) and recruitment of these neurons lead to an impairment and enhancement of exercise capacity, respectively. The study is interesting in that it provides support to a role of the dorsal vagal motor nucleus (DVMN) in exercise capacity. The investigators have used state of the art experimental tools (genetic silencing via lentiviral infections of allatostatin receptor, optogenetic stimulation) to generate data in support to their hypothesis. Thus even though the conceptual underpinning of the study is not particularly novel (vagal tone affects exercise capacity), the data generated do provide evidence for DVMN neurons involvement in exercise. Nevertheless, there are a number of general and specific concerns that significantly reduce this reviewer's enthusiasm for the overall study. First, the rationale and development of the work are unclear to this reviewer. Apparently, the study was designed to directly test if the strength of vagal tone determines exercise capacity. However, the authors only tested the role of a specific group of neurons (dorsal vagal motor nucleus-DVMN) of the vagus nerve and thus they cannot extrapolate to what happened with the systemic vagal stimulation. In my opinion such sentences stating that "the results demonstrate the involvement of vagal activity on the exercise" (e.g. the last sentence of the abstract or even the title of the paper) should be tuned down as they are limited only to DVMN. It is known that physical exercise increases sympathetic activity and decreases parasympathetic activity. Upon cessation of exercise, parasympathetic activity reactivates. Systemic deactivation/reactivation of the parasympathetic activity could modify responses and alter effects observed by the authors. Second, do the authors have any direct evidence of decrease of parasympathetic activity after expression of AlstR and treatment with allatostatin in DVMN neurons? The same would apply to the recruitment protocol. If so, it should be mentioned. Third in my opinion the analysis of the mechanism underlying the relationship is missing. How silencing or activation of DVMN activity can lead to an increase in exercise capacity? In fact, the authors should also discuss in brief the possible mechanisms by which vagal input affects the exercise capacity. Is that achieved at the level of the locomotor system, cardiac performance, central nervous system activity...? Unfortunately the authors do not go deep to show a mechanism of how the activation DVMN improves the function of the heart after endurance exercise training.

Specific comments.

1. Figure 1a. Authors stated that there is a relationship between DVMN firing frequency and 24 h distance measured in mice which supports the hypothesis that the intrinsic vagal activity may determine the ability to exercise (end of page 2). This graph is somewhat misleading, however. It seems that 24 h distance increased between ≈ 0.5 and ≈ 0.9 Hz, was stable between ≈ 0.9 and ≈ 1.4 Hz, and steeply increased between ≈ 1.4 and 1.7 Hz. Surprisingly, at firing frequencies >1.7 Hz, distance substantially decreased. Unfortunately, this means that there is no clear dependency and, thus, these results do not support the conclusion made by the authors. Do the authors have an explanation to what happened at frequencies >1.7 Hz?

2. As stated above, the relationship between the DVMN firing frequency and exercise distance in Figure 1a is not linear but rather biphasic. In contrast, the other experiments suggest a somewhat linear relationship between vagal input and exercise capacity. This raises the issue of the specificity in either (i) the DVMN recordings (can they be contaminated with sympathetic input?), or (ii) whether the manipulations also involve some other pathways? Further, it would have been good to see the DVMN recordings in brain slices after the manipulations (silencing and enhancement).

3. Again regarding Figure 1a, the fact that the authors used cell-attached recordings precludes one

from knowing the specific resting membrane potential (RMPs) of the neurons from which the records were taken. Clearly, firing frequency depends on the RMP. Since no composite data from multiple recordings are presented from each condition one cannot be sure that the recordings presented are simply coincidental from neurons that had different RMPs.

3. Figure 1d. In AlstR animals DVMN silencing did not seem to significantly modify mean arterial pressure (MAP). This would be consistent with the scarce role of the parasympathetic nervous system in the control of basal vascular tone. However, it is surprising the significant reduction of the MAP increase on peak exercise as compared with control animals. What is the mechanism underlying this effect? On the other hand, in these animals heart rate was not increased compared to baseline either. This would imply that, as suggested by the authors in other papers, DVMN neurons would innervate mainly the ventricles but not the sinus node. This result also prevents the extrapolation of the present results to the general parasympathetic nervous system, whose suppression does have effects on heart rate.

4. Another major concern relates to the relevance of the rodent model to humans: if one looks at the heart rate recovery in Fig. 1d, the curves are very similar for eGFP and AlstR cases; this is in contrast to the one shown in 2F, where a slower recovery is shown as a hallmark of vagal dysfunction in humans. Thus one wonders how robustly the investigators are mimicking vagal dysfunction in their experiments/animal models, or alternately, whether heart rate recovery is a good surrogate for the same.

5. The authors reported the absence of changes in the chronotropic effect of dobutamine in ChIEF expressing animals, but did not show the results obtained in silenced animals. If DVMN silencing does not modify the chronotropic effect of dobutamine, what is the reason for the significant reduction of the heart rate compared to control animals at peak exercise?

6. Figure 1e and Figure 2d. The rationale of the experiments analyzing the effects of DVMN silencing or activation on the cardiac responses to beta1 adrenergic stimulation is not mentioned. In fact, the use of beta1 adrenergic stimulation to test for neurally mediated responses in heart rate or contractility does not make any sense since dobutamine acts by directly combining with beta1 adrenergic receptors at the cardiac myocyte membrane. How is it therefore that in experiments shown in Figure 1e show allatostatin treatment in animals expressing AlstR completely blunted the inotropic response to beta1-adrenoceptor stimulation with dobutamine, or that DVMN activation increased baseline ejection fraction and enhanced contractile response to dobutamine. What is the mechanism underlying this effect? Is altered sensitivity to catecholamines a consequence of DVMN silencing/activation or the cause of the reduced/enhanced exercise capacity? In activation experiments, a third group of animals was added (naïve animals with training). Authors should clarify why they added this group only as comparator to activation of DVMN and not to silencing. Since the nature of the experiments and protocols used in silencing and activating groups are different, the interpretation of the results is somewhat difficult.

7. Figure 2f-I (human participants). The large number of participants studied is valuable but we are never told of the clinical characteristics of the participants or the medications they were taking, particularly those individuals diagnosed with parasympathetic dysfunction.

8. The Authors stratified subjects on the basis of the HRR and considered the vagal dysfunction group those with HRR below 12 bpm. However, this is an indirect measure of vagal activity and other reasons for decreased heart rate recovery (cardiac and noncardiac diseases, drugs, etc) seem not to be considered. Figure 2i was not mentioned by the authors in the text. Do the authors observe differences on any other parameters measuring exercise capacity (time to fatigue, differences in maximal tolerance) apart from those already given?

Reviewer #2 (expert in neural regulation of the heart)

Remarks to the Author:

The findings reported in this brief communication provide an important contribution to the knowledge of the relationship between vagal tone and exercise capacity. Despite the large volume of studies showing correlations between parasympathetic activity and endurance capacity, no previous studies addressed the key unanswered question of whether or not increased brainstem vagal activity contributes in a causal manner to increased endurance capacity. The authors use optogenetic methods to selectively control activity of cardiovagal neurons in the dorsal motor nucleus of the vagus to show that reduced DMNV neuron activity greatly reduced exercise capacity. Conversely, stimulation of DMNV neurons increased exercise capacity. Additional experiments showed that reduced DVMN activity impairs the ability of the heart to respond to sympathetic stimulation. A very intriguing finding.

The results and their interpretation are clear.

One concern/suggestion is that it would be informative to include data on heart rate variability given the clinical significance of changes in heart rate variability.

Reviewer #3 (expert in optogenetics and neural regulation of heart function)

Remarks to the Author:

This is an interesting manuscript that presents new data from four different experiments to elucidate the effect of cardiac vagal tone on exercise capacity in rats. State-of-the-art experimental approaches were used to both reduce and elevate DVMN neuron firing rate during 6-day exercise protocols. Overall, the experimental design consisting of "control", DVMN activated or inhibited, and exercise-trained animals is relatively robust.

The primary limitation of the work is associated with the perceived non-specificity of the genetic targeting of the DVMN vagal pre-ganglionic neurons. Although vagal preganglionic neurons of the DVMN express the transcriptional factor Phox2, this targeting is unfortunately non-specific. Many other cell types and neurons, including the NTS neurons that are neighbors to DVMN neurons, also express Phox2 (see Kang and colleagues *J. Comp. Neurol.* 503(5)627-641, 2007). This non-selective approach diminishes the impact and interpretation of the results. It is possible, if not likely, that many of the observed effects are due to changes in the NTS sensory autonomic nucleus. Chemoreceptor and pain pathways were likely inadvertently altered as well using these non-selective approaches.

What percentage of the DVMN neurons that were targeted and/or studied project to the heart? The vast majority of DVMN neurons project to visceral targets within the respiratory system and gastrointestinal track, in addition to the minority that project to the heart. Since the experiments were focused upon cardiac function an assessment of the DVMN neurons that project to the heart is likely to be required.

Results shown in Figs 1e and supplemental Fig 1 require additional explanation. Upon studying these figures, it appears that DVMN silencing provides almost complete beta-adrenergic activation before the addition of dobutamine (Supp Fig 1). LVESP is approx 165mmHg at paced baseline with allatostatin and is almost the same without allatostatin but with dobutamine. A similar result was observed in the contractility measurements (Fig 1e). LV ejection fraction increased with DVMN activation, presumably due to increased relaxation and increased EDV? How did contractility change during DVMN activation?

It would be informative for the authors to present a working mechanistic theory and explain the primary signaling pathways that motivate such powerful vagal modulation of cardiac reactivity to catecholamines, as stated in paragraph #6. Specific physiological mechanisms for the interaction

between vagal tone and beta-receptor activation are not discussed and would enrich the manuscript.

- A. Originality and interest: the studies are novel and should be of interest to a wide audience.
- B. Data & methodology: The primary limitation is the perceived non-specificity of the genetic targeting of the DVMN vagal pre-ganglionic neurons. This must be addressed.
- C. Appropriate use of statistics and treatment of uncertainties: Statistics were appropriate.
- D. Conclusions: robustness, validity, reliability: Reliability could be tainted by the non-specificity limitation.
- E. Suggested improvements: Other suggestions provided above.
- F. References: References are appropriate.
- G. Clarity and context: The text is clearly written and the figures nicely present the data.

MS ID: NCOMMS-16-14931-T
Responses to referees' comments

We are grateful for the constructive comments of three reviewers of *Nature Communications* and have taken full account of the raised criticisms. With this appeal, we now include additional experimental data requested by the reviewers, provide a full response to their comments and submit a thoroughly revised manuscript (with all the major changes **highlighted**). Below we state the criticisms ("critique") and then provide our responses.

Reviewer #1 (expert in neural regulation of the heart)

General comments

The stated objective of this study was to evaluate the involvement of cardiac vagal activity in the exercise capacity. The results showed that in experimental models, silencing of brainstem vagal preganglionic neurons (dorsal vagal motor nucleus-DVMN) and recruitment of these neurons lead to an impairment and enhancement of exercise capacity, respectively. The study is interesting in that it provides support to a role of the dorsal vagal motor nucleus (DVMN) in exercise capacity. The investigators have used state of the art experimental tools (genetic silencing via lentiviral infections of allatostatin receptor, optogenetic stimulation) to generate data in support to their hypothesis.

Response: We would like to thank this referee for his/her time taken to review our manuscript and overall positive assessment of our work. We now include additional experimental data, provide our responses to all the criticisms raised and submit a thoroughly revised manuscript.

Critique: Thus even though the conceptual underpinning of the study is not particularly novel (vagal tone affects exercise capacity), the data generated do provide evidence for DVMN neurons involvement in exercise.

Response: We respectfully disagree with the reviewer here. There is no direct experimental evidence demonstrating a *causal* link between the strength of parasympathetic (vagal) tone and exercise capacity. Indirect measures of cardiac vagal activity are indeed strongly associated with exercise capacity, but this association is difficult to dissect. This association may be attributable to vagal tone being enhanced by exercise training (an issue currently contentiously debated, see for example *J Physiol* 593: 1745, 2015; opposing view in *Nat Commun* 5:3775, 2014) or, alternatively, intrinsically higher parasympathetic activity promoting higher tolerance for endurance exercise training. In order to demonstrate *causality*, this study directly examined how exercise capacity is affected in conditions of experimentally-induced enhancement or suppression of vagal tone.

Critique: Nevertheless, there a number of general and specific concerns that significantly reduce this reviewer's enthusiasm for the overall study. First, the rationale and development of the work are unclear to this reviewer.

Response: Please see the response to the above comment. This study was designed to directly test the hypothesis that the strength of vagal tone determines exercise capacity, implying that vagal withdrawal should reduce while vagal recruitment should enhance the ability to exercise. We believe that addressing this question is fundamentally important for our understanding of exercise physiology in general and the role of the autonomic nervous system in determining exercise capacity in particular. For example,

please see the debate on this matter published last year in *The Journal of Physiology (J Physiol 593: 1745, 2015)*.

Critique: Apparently, the study was designed to directly test if the strength of vagal tone determines exercise capacity. However, the authors only tested the role of a specific group of neurons (dorsal vagal motor nucleus-DVMN) of the vagus nerve and thus they cannot extrapolate to what happened with the systemic vagal stimulation. In my opinion such sentences stating that "the results demonstrate the involvement of vagal activity on the exercise" (e.g. the last sentence of the abstract or even the title of the paper) should be tuned down as they are limited only to DVMN.

Response: We respectfully disagree with the reviewer here. Vagal motor activity is provided by groups of vagal preganglionic neurons residing within two discrete brainstem nuclei - the nucleus ambiguus (NA) and the dorsal vagal motor nucleus (DVMN). DVMN neurons generate the majority of central parasympathetic activity, while NA neurons appear to provide rhythmic chronotropic vagal tone via innervation of the cardiac nodal tissue. NA neurons are rhythmic, while DVMN neurons are tonic. As we discussed in detail in our recent review on cardiac vagal preganglionic neurons (Gourine et al *Auton Neurosci.* 199:24, 2016), studies of the ontogenesis of the CNS vagal system in metamorphosing amphibians undergoing anatomical and physiological changes during the transition from water- to air breathing indicate that the DVMN is the primary vagal nucleus. During metamorphosis, rapidly developing cardiorespiratory interactions initiate ventral migration of a subset of DVMN neurons giving rise to a compact formation of the NA which acquires respiratory modulation of activity from the neighbouring respiratory network. Our recent work demonstrated that the electrical and contractile properties of the left ventricle of the heart are controlled by the neuronal projections originating from the DVMN (Machhada et al *Heart Rhythm* 12, 2285, 2015; Machhada et al. *J Physiol.* 594: 4017, 2016). It appears that vagal innervation of the left ventricle is provided predominantly by the DVMN neuronal projections which (as our new data suggest) are critically important in determining the responsiveness of ventricular myocytes to β -adrenoceptor activation (please see below). In the revised manuscript we now give justification of why this work is focused on the DVMN neurons.

Critique: It is known that physical exercise increases sympathetic activity and decreases parasympathetic activity. Upon cessation of exercise, parasympathetic activity reactivates. Systemic deactivation/reactivation of the parasympathetic activity could modify responses and alter effects observed by the authors.

Response: We certainly agree with the reviewer here, yet withdrawal and reactivation of parasympathetic activity during and after the exercise may only be relevant for the chronotropic control of the nodal tissue. Ventricular innervation by the DVMN neuronal projections is tonic and we do not really know if DVMN neurons exhibit the same pattern of exercise-induced changes in activity.

Critique: Second, do the authors have any direct evidence of decrease of parasympathetic activity after expression of AlstR and treatment with allatostatin in DVMN neurons? The same would apply to the recruitment protocol. If so, it should be mentioned.

Response: We indeed mention this in the original version of our manuscript (Page 2: "Insect peptide allatostatin rapidly inhibits autonomic neurons transduced to express AlstR^{13,14} while 445 nm light pulses trigger precisely timed depolarizations and action potential firing of DVMN neurons expressing ChIEF¹⁴"). Full characterization of the DVMN silencing using AlstR/allatostatin approach and DVMN activation using light stimulation of

ChIEF is given in our earlier publication (*Cardiovasc Res* 95, 487, 2012) and below we reproduce published illustrations showing: **(1A)** a representative example of distribution of choline acetyltransferase (ChAT)-positive (i.e. cholinergic) (red) DVMN neurons transduced to express AlstR/eGFP (green). XII – hypoglossal motor nucleus, ChAT-positive, but not expressing AlstR/eGFP; **(1B)** A representative example of distribution of AlstR/eGFP-transduced DVMN neurons in relation to the location of A2 noradrenergic cells identified by tyrosine hydroxylase (TH) immunohistochemistry (red). **(1C)** Representative cell-attached recording from an AlstR/eGFP-positive DVMN neuron illustrating its rapid and reversible silencing in response to allatostatin (0.5 μM); **(1D)** Current-voltage relationship (IV) of allatostatin-induced current. Transduced DVMN neuron was voltage-clamped to -30 mV and hyperpolarizing voltage ramps to -130 mV (700 ms duration) were applied before and during allatostatin application. The displayed IV was obtained by subtracting the whole-cell IV obtained under control conditions, from that obtained in the presence of allatostatin. **(2A)** Representative whole-cell current-clamp recording from ChIEFtdTomato-expressing DVMN neuron illustrating depolarization and action potential firing in response to blue light (20 ms pulses). Four consecutive traces are overlaid. Action potential was elicited in response to 15 out of 16 pulses; **(2B)** DVMN neurons expressing ChIEFtdTomato. Scale bar=30 μm **(2C)** Mean data from voltage-clamp recordings (n=6) at a holding potential of -50 mV demonstrating the relationship between duration of the light stimulus and the normalized amplitude of inward current elicited by opening ChIEF channel.

Figure 1

Figure 2

Critique: Third in my opinion the analysis of the mechanism underlying the relationship is missing. How silencing or activation of DVMN activity can lead to an increase in exercise capacity? In fact, the authors should also discuss in brief the possible mechanisms by which vagal input affects the exercise capacity. Is that achieved at the level of the locomotor system, cardiac performance, central nervous system activity...? Unfortunately the authors do not go deep to show a mechanism of how the activation DVMN improves the function of the heart after endurance exercise training.

Response: We fully agree with the reviewer here and for this resubmission experimentally explored one of the potential mechanisms. Enhanced cardiac contractility (augmented inotropic state) following optogenetic recruitment of the DVMN neuronal projections suggested that the heart became more responsive to β -adrenoceptor stimulation. Therefore, we compared the level of expression of G protein-coupled receptor kinase 2 (GRK2) and β -Arrestin 2 in left ventricular myocytes of rats subjected to four daily sessions of vagal stimulation by optogenetic recruitment of the DVMN activity (DVMN neurons transduced to express light-sensitive channel ChIEF) and control animals (expressing eGFP) which received sham stimulation with blue-light illumination. GRKs and arrestins are key negative regulators of GPCR (including β -adrenoceptor)-mediated signalling (*Mol Pharmacol*, 63, 9, 2003). GRKs promote phosphorylation of the intracellular domain of the active receptor, recruiting arrestins to block GPCR coupling to G proteins resulting in receptor desensitization and internalization. In this revised submission we now include the new data (Figure 2f) which demonstrate that optogenetic stimulation of the DVMN markedly reduces the level of both GRK2 and β -Arrestin 2 expression in the left ventricle. From these data we conclude that DVMN neuronal projections to the left ventricle control the expression of GRK2 and β -Arrestin 2, and, therefore, modulate responsiveness of cardiomyocytes to β -adrenoceptor stimulation, control ventricular contractility and determine the exercise capacity. This conclusion is also supported by the results of our study published earlier this year which used a rat model of baroreflex dysfunction and demonstrated a clear association between parasympathetic dysfunction, impaired cardiac contractility and increased left ventricular GRK2 expression (Ackland et al. *Crit Care Med* 44: e614, 2016), although exercise capacity was not assessed. This notion is also fully consistent with the established role of GRK2 in the heart (*Mol Pharmacol*, 63, 9, 2003) and data on cross-talk between muscarinic and adrenoceptor-mediated signalling (see for example *Mol Pharmacol* 56, 813, 1999).

Specific comments.

Critique: Figure 1a. Authors stated that there is a relationship between DVMN firing frequency and 24 h distance measured in mice which supports the hypothesis that the intrinsic vagal activity may determine the ability to exercise (end of page 2). This graph is somewhat misleading, however. It seems that 24 h distance increased between ≈ 0.5 and ≈ 0.9 Hz, was stable between ≈ 0.9 and ≈ 1.4 Hz, and steeply increased between ≈ 1.4 and 1.7 Hz. Surprisingly, at firing frequencies >1.7 Hz, distance substantially decreased. Unfortunately, this means that there is no a clear dependency and, thus, these results do not support the conclusion made by the authors. Do the authors have an explanation to what happened at frequencies >1.7 Hz?

Response: Figure 1a summarizes the data obtained from the recordings of 115 DVMN neurons *in vitro* (i.e. in the absence of afferent modulation from the periphery and the rest of the CNS) from 14 mice (8 cells on average). The data show that mean DVMN neuronal action potential firing rate in the range 0-1.7 Hz increases with the amount of voluntary exercise performed by the animals in a 24 h period. We agree with the reviewer that the data also demonstrate that DVMN firing frequency above 2Hz

(recorded in the brainstem slices *in vitro*) is associated with reduced voluntary exercise. The shape of this relationship is similar to that of many curves one may find in every good Physiology textbook. We hypothesize that within a lower range of *in vitro* firing frequencies (as our data suggest between 0 and 1.7 Hz in mice) there is a strong relationship between the DVMN activity and the amount of voluntary exercise performed by the animals. Increasing levels of the DVMN neuronal activity are hypothesized to result in lower levels of GRK2/arrestin expression in cardiac myocytes as discussed in detail above. However, very high resting DVMN activity may limit cardiac contractility via direct actions of acetylcholine on ventricular myocytes (as we demonstrated in our earlier publication, Machhada et al. *J Physiol.* 594: 4017, 2016). We now discuss this in the revised version of the manuscript.

Critique: As stated above, the relationship between the DVMN firing frequency and exercise distance in Figure 1a is not linear but rather biphasic. In contrast, the other experiments suggest a somewhat linear relationship between vagal input and exercise capacity. This raises the issue of the specificity in either (i) the DVMN recordings (can they be contaminated with sympathetic input?), or (ii) whether the manipulations also involve some other pathways? Further, it would have been good to see the DVMN recordings in brain slices after the manipulations (silencing and enhancement).

Response: The only purpose of this experiment was to assess the relationship between the DVMN neuronal activity and the amount of voluntary exercise performed by the study animals. We recorded the activities of 115 neurons in 14 individual mice; these data suggest that within the lower range of DVMN frequencies there is a relationship between the DVMN firing frequency and distance covered by the animals in a 24h period (please see our response to the previous comment). (i) The recordings were performed *in vitro* using brainstem slice preparations with visual identification of DVMN neurons based on anatomical location and distinctive appearance. Therefore, these recordings are unlikely to be contaminated by the sympathetic input; (ii) It is unlikely that our manipulations involve other pathways (in addition to inhibition or recruitment of the DVMN neuronal activity) as detailed in our response to the third reviewer; (iii) We respectfully disagree with the reviewer that the DVMN recordings in brain slices after the manipulations would provide useful information. As described in detail above, the validity of our approach in silencing and activation of the DVMN neurons was demonstrated previously. Also, both of these treatments are readily reversible, therefore, it is highly unlikely that the differences in resting DVMN activity will be observed following isolation of the DVMN in a slice preparation.

Critique: Again regarding Figure 1a, the fact that the authors used cell-attached recordings precludes one from knowing the specific resting membrane potential (RMPs) of the neurons from which the records were taken. Clearly, firing frequency depends on the RMP. Since no composite data from multiple recordings are presented from each condition one cannot be sure that the recordings presented are simply coincidental from neurons that had different RMPs.

Response: The firing frequency depends on the RMP and on the input resistance as well as any synaptic inputs. The combination of these three parameters determines whether at any given point in time the cell fires an action potential or not. For the purpose of this study we have recorded DVMN neuronal activity *in vitro* in isolated 200 μm thick brainstem slices. It was important to maintain the recording and 'culturing' conditions identical for all the animals used. It also meant that the majority of afferent inputs are removed and we are recording an 'intrinsic' activity that is only modulated by spontaneous synaptic inputs from adjacent cells in the same slice preparation. Our aim was to influence this intrinsic activity as little as possible with our recording conditions.

This prompted us to opt for the 'least invasive' form of patch clamp recordings: the loose, attached patch. This configuration only provides information about the firing rate, but exactly this is the single most important parameter we need to record in this case. Whilst this recording configuration does not provide us with information about the RPM, it also precludes us from influencing RPM with the composition of the patch pipette solution.

DVMN contains a heterogeneous population of neurons regarding their projection targets and in a slice preparation there is no information available on the target of each individual neuron recorded. Consequently, the recordings presented here for each of the animals represent a random selection of DVMN neurons. Given the random nature of sampling, we were surprised by how clear this association between the animals exercise history and vagal activity presented itself. The experiments conducted here have simply recorded this association, but do not in itself indicate any causal relationship. It is not clear to us what point the reviewer is making when stating 'that the recordings presented are simply coincidental from neurons that had different RMPs', because this is the purpose of this study to get a representation of the population of DVMN neurons from each of the animals without manipulating their properties or preselecting for a given membrane potential.

Critique: Figure 1d. In AlsR animals DVMN silencing did not seem to significantly modify mean arterial pressure (MAP). This would be consistent with the scarce role of the parasympathetic nervous system in the control of basal vascular tone. However, it is surprising the significant reduction of the MAP increase on peak exercise as compared with control animals. What is the mechanism underlying this effect?

Response: Significantly smaller increases in mean arterial blood pressure during exercise in conditions of DVMN silencing are consistent with the rest of the data obtained in this study. Figure 1e shows that in the absence of the DVMN input the heart is not able to increase the force of left ventricular contraction in response to β -adrenoceptor stimulation.

Critique: On the other hand, in these animals heart rate was not increased compared to baseline either. This would imply that, as suggested by the authors in other papers, DVMN neurons would innervate mainly the ventricles but not the sinus node. This result also prevents the extrapolation of the present results to the general parasympathetic nervous system, whose suppression does have effects on heart rate.

Response: We respectfully disagree with the reviewer here. Figure 1d (right panel) clearly shows a marked increase (albeit slightly smaller than in the control animals) in heart rate during exercise in conditions of DVMN silencing. Please see our responses to the above comments which discuss the relative contribution of NA and DVMN neurons to the generation of the efferent vagal tone and reasons why this study focused on the DVMN neuronal population.

Critique: Another major concern relates to the relevance of the rodent model to humans: if one looks at the heart rate recovery in Fig. 1d, the curves are very similar for eGFP and AlstR cases; this is in contrast to the one shown in 2F, where a slower recovery is shown as a hallmark of vagal dysfunction in humans. Thus one wonders how robustly the investigators are mimicking vagal dysfunction in their experiments/animal models, or alternately, whether heart rate recovery is a good surrogate for the same.

Response: In humans, the speed of heart rate recovery upon cessation of exercise provides the best estimate of individual ability to recruit chronotropic vagal tone. For the

human study, we make an assumption that the strength of chronotropic vagal tone (i.e. vagal influence on the nodal tissue) correlates with the strength of parasympathetic outflow to the left ventricle. Experimental animal studies allow us to switch “on” and “off” distinct populations of vagal preganglionic neurons controlling different aspects of heart physiology and this study focused on the DVMN population which provides functional vagal innervation of the left ventricle (Machhada et al *Heart Rhythm* 12, 2285, 2015; Machhada et al. *J Physiol.* 594: 4017, 2016). In the revised manuscript we now clearly indicate why our experimental animal work is focused specifically on the DVMN neurons which innervate the left ventricle and have limited control over the chronotropic function. We also acknowledge that “*Although the strength of parasympathetic tone to the left ventricle in humans is impossible to measure directly, the rate of heart rate recovery (HRR) after cessation of exercise serves as a robust index of individual ability to recruit vagal tone (Figure 3a) and is highly sensitive to muscarinic blockade (Imai et al., 1994). Assessment of HRR avoids deriving parasympathetic activity from measures dependent on absolute heart rate – a major confounder in heart rate variability analysis (Monfredi et al., 2014)*”.

Critique: The authors reported the absence of changes in the chronotropic effect of dobutamine in ChiEF expressing animals, but did not show the results obtained in silenced animals.

Response: In our earlier detailed study we evaluated the effect of DVMN silencing on heart rate and ventricular contractility in a rat model (Machhada et al. *J Physiol.* 594: 4017, 2016). We reported that inhibition of DVMN using this approach had no effect on heart rate which is consistent with differential innervation of the nodal tissue and ventricular myocardium by the NA and DVMN neuronal projections (as discussed in detail above; please also see our recent review article on cardiac vagal preganglionic neurons [Gourine et al *Auton Neurosci.* 199:24, 2016]). However, since the heart rate fluctuates during the course of long protocols, in the experiment illustrated on Figure 1e, we ensured constant heart rate conditions before and after administration of allatostatin by pacing the heart at 20% above the initial resting heart rate (described in the Methods and also indicated on the Figure). Experiment illustrated on Figure 2e involved a much shorter protocol, therefore, the heart was not paced, which also allowed us to determine the effect of treatment on dobutamine-induced changes in heart rate.

Critique: If DVMN silencing does not modify the chronotropic effect of dobutamine, what is the reason for the significant reduction of the heart rate compared to control animals at peak exercise?

Response: In the control animals, the heart rate increased during exercise to ~545 bpm, while in conditions of DVMN silencing the heart rate increased to ~510 bpm. Although this difference is indeed statistically significant, it is rather small (<10%) and may simply reflect the fact that experimental animals were unable to reach the same level of exercise performance when compared to the controls.

Critique: Figure 1e and Figure 2d. The rationale of the experiments analyzing the effects of DVMN silencing or activation on the cardiac responses to beta1 adrenergic stimulation is not mentioned. In fact, the use of beta1 adrenergic stimulation to test for neurally mediated responses in heart rate or contractility does not make any sense since dobutamine acts by directly combining with beta1 adrenergic receptors at the cardiac myocyte membrane. How is it therefore that in experiments shown in Figure 1e show allatostatin treatment in animals expressing AlstR completely blunted the inotropic response to beta1-adrenoceptor stimulation with dobutamine, or that DVMN activation increased baseline ejection fraction and enhanced contractile response to dobutamine.

What is the mechanism underlying this effect? Is altered sensitivity to catecholamines a consequence of DVMN silencing/activation or the cause of the reduced/enhanced exercise capacity?

Response: We thank the reviewer for this comment and for this resubmission performed additional experiments designed to explore the potential underlying mechanism(s). In the revised manuscript we now include the new data (Figure 2f) showing that optogenetic stimulation of the DVMN activity markedly reduces GRK2/ β -Arrestin 2 expression in left ventricular myocytes. From these data we conclude that DVMN neuronal projections to the left ventricle control the expression of GRK2 and arrestins, and, therefore, modulate responsiveness of cardiomyocytes to β -adrenoceptor stimulation, control ventricular contractility and determine the exercise capacity. We now revised the text of the manuscript to provide a clear rationale of doing these experiments: *"We next determined whether reduced activity of the DVMN neurons may alter cardiac contractile responses to sympathetic β -adrenoceptor-mediated stimulation, which is essential to trigger and maintain appropriate increases in cardiac output to support circulatory requirements of exercise"*.

Critique: In activation experiments, a third group of animals was added (naïve animals with training). Authors should clarify why they added this group only as comparator to activation of DVMN and not to silencing. Since the nature of the experiments and protocols used in silencing and activating groups are different, the interpretation of the results is somewhat difficult.

Response: We respectfully disagree with the reviewer here. Addition of the third group of animals in the design of the experiment involving optogenetic stimulation of the DVMN neurons is logical and allows comparison of the effects achieved by the experimental treatment (DVMN recruitment) and that of exercise training. Inclusion of an analogous group in the experiment involving acute DVMN silencing is problematic as this would require some form of natural "detraining" of rats showing reasonable exercise capacity. This can be done in humans, but not in young normally behaving experimental animals.

Critique: Figure 2f-I (human participants). The large number of participants studied is valuable but we are never told of the clinical characteristics of the participants or the medications they were taking, particularly those individuals diagnosed with parasympathetic dysfunction.

Response: We thank the reviewer for acknowledging the value of this experiment which provides strong support of the conclusions derived from the experimental animal studies. We now include the requested information in the revised version of the manuscript (Supplementary Tables 1 and 2). Patients with/without delayed HRR had similar preoperative cardiovascular, renal and nutritional (as reflected by albumin) profiles. Diabetic patients comprised 19.8% of the population with delayed HRR. Patients with delayed HRR were more likely to receive cardiovascular (relative risk:2.18 (95 per cent c.i. 1.80-2.63); $p<0.01$) and diabetic medications (relative risk:1.60 (95 per cent c.i. 1.18-2.17); $p<0.01$). On multivariate logistic regression analysis, the only factors associated with delayed HRR were age and diabetes mellitus. Multivariable logistic regression analysis showed no association with any cardiovascular medication (including β -blockade).

Critique: The Authors stratified subjects on the basis of the HRR and considered the vagal dysfunction group those with HRR below 12 bpm. However, this is an indirect measure of vagal activity and other reasons for decreased heart rate recovery (cardiac and noncardiac diseases, drugs, etc) seem not to be considered. Figure 2i was not

mentioned by the authors in the text. Do the authors observe differences on any other parameters measuring exercise capacity (time to fatigue, differences in maximal tolerance) apart from those already given?

Response: We thank the reviewer for this comment. Please see our response to the previous comment. The exercise protocol was fatigue/symptom limited, so objective measures at this point were reported.

Reviewer #2 (expert in neural regulation of the heart)

The findings reported in this brief communication provide an important contribution to the knowledge of the relationship between vagal tone and exercise capacity. Despite the large volume of studies showing correlations between parasympathetic activity and endurance capacity, no previous studies addressed the key unanswered question of whether or not increased brainstem vagal activity contributes in a causal manner to increased endurance capacity. The authors use optogenetic methods to selectively control activity of cardiovagal neurons in the dorsal motor nucleus of the vagus to show that reduced DMNV neuron activity greatly reduced exercise capacity. Conversely, stimulation of DMNV neurons increased exercise capacity. Additional experiments showed that reduced DVMN activity impairs the ability of the heart to respond to sympathetic stimulation. A very intriguing finding. The results and their interpretation are clear.

Response: We would like to thank this referee for their extremely positive assessment of our work.

Critique: One concern/suggestion is that it would be informative to include data on heart rate variability given the clinical significance of changes in heart rate variability.

Response: We thank the reviewer for this comment but believe very strongly that the speed of heart rate recovery upon cessation of exercise provides the best possible measure of individual ability to recruit vagal tone. There is a strong argument that heart rate variability is primarily dependent on the prevailing heart rate and cannot be used in any simple way to assess autonomic control of the heart (*Hypertension* 64, 1334, 2014). We are not in a position to contribute to this debate (it would also be beyond the scope of our report) and in this study used heart rate recovery as a surrogate measure of parasympathetic (dys)function.

Reviewer #3 (expert in optogenetics and neural regulation of heart function)

This is an interesting manuscript that presents new data from four different experiments to elucidate the effect of cardiac vagal tone on exercise capacity in rats. State-of-the-art experimental approaches were used to both reduce and elevate DVMN neuron firing rate during 6-day exercise protocols. Overall, the experimental design consisting of "control", DVMN activated or inhibited, and exercise-trained animals is relatively robust.

Response: We would like to thank this referee for his/her time taken to review our manuscript and overall positive assessment of our work. We now include additional experimental data, provide our responses to all the criticisms raised and submit a thoroughly revised manuscript.

Critique: The primary limitation of the work is associated with the perceived non-specificity of the genetic targeting of the DVMN vagal pre-ganglionic neurons. Although vagal preganglionic neurons of the DVMN express the transcriptional factor Phox2, this targeting is unfortunately non-specific. Many other cell types and neurons, including the NTS neurons that are neighbors to DVMN neurons, also express Phox2 (see Kang and colleagues *J. Comp. Neurol.* 503(5)627-641, 2007). This non-selective approach diminishes the impact and interpretation of the results. It is possible, if not likely, that many of the observed effects are due to changes in the NTS sensory autonomic nucleus. Chemoreceptor and pain pathways were likely inadvertently altered as well using these non-selective approaches.

Response: We respectfully disagree with the reviewer here. In our study DVMN was targeted using viral vectors driving the expression of active and control transgenes under the control of the PRSx8 promoter – Phox2 activated promoter. Our group was, in fact, the first (after its original description by SK-Kim's group) to use PRSx8 promoter (*Physiol Genomics* 20:165, 2005) to target autonomic neurons and we have an extensive experience with it. Surprisingly, not all cells which express Phox2 efficiently drive PRSx8 promoter, hence the presence of this transcriptional factor as such is not yet a guarantee for the appearance of the designed transgenes. In the dorsal brainstem only DVMN and A2 neurons of the neighbouring NTS express Phox2 transcription factors, therefore PRSx8 promoter can only be active in these two populations of cells. A2 neurons can in principle express transgenes driven by PRSx8. This was demonstrated in our earlier studies (*Cardiovasc Res* 76: 184, 2007) by injecting adenoviral vectors directly into the NTS. Already in these early experiments we noticed that the DVMN neurons are much more sensitive to PRSx8-bearing vectors. When PRSx8-driven constructs are placed in the lentiviral vectors, as in the present work, and injections are made in the DVMN or directly below its anatomical boundary, expression is essentially completely selective to the DVMN. This is easy to see since A2 neurons project rostrally and do not overlap with the DVMN population.

In our more recent publications we fully characterized specificity of viral targeting of the DVMN vagal preganglionic neurons using the vectors and approach used in this study (*Cardiovasc Res* 95: 487, 2012; *Heart Rhythm* 12: 2285, 2015; *J Physiol* 594: 4017, 2016). Please review **Figure 1** (taken from *Cardiovasc Res* 95: 487, 2012) above which is provided here in response to one of the comments raised by the first reviewer. **Panel 1A** demonstrates that the vast majority (if not all) of dorsal brainstem neurons transduced to express AlstR/eGFP express choline acetyltransferase (ChAT), i.e. these cells are cholinergic neurons. In this area of the brainstem only DVMN and hypoglossal motoneurons are cholinergic. Hypoglossal neurons (XII nucleus) were indeed identified by ChAT immunoreactivity, but were not transduced as they do not express Phox2. **Panel 1B** shows a representative example of distribution of AlstR/eGFP-transduced DVMN neurons in relation to the location of A2 noradrenergic cells identified by tyrosine hydroxylase (TH) immunohistochemistry. Along with strong expression of AlstR/eGFP in the DVMN only occasional noradrenergic neurons were found to be transduced.

In our latest studies (*Heart Rhythm* 12: 2285, 2015; *J Physiol* 594: 4017, 2016) we further refined DVMN targeting by placing viral microinjections 0.1-0.2 mm ventral to the anatomical boundary of the DVMN (we now indicate this in the revised version of the manuscript). This limited/prevented diffusion of viral particles to the NTS, completely avoiding potential transduction of the A2 neurons, while sparing hypoglossal motoneurons. Please review the images below; **Figure 3** is taken from our recently published work (*J Physiol* 594: 4017, 2016) and **Figure 4** is provided in this submission. Both clearly show that the expression of the transgenes is restricted to the DVMN with no specific labelling observed outside the anatomical boundaries of the nucleus.

Figure 3

A, Photomicrographs of the coronal sections of the rat brainstem showing expression of eGFP (amplified by immunohistochemistry) in a control animal injected with LVV-PRsX8-eGFP. Images illustrate representative example of the distribution of transduced DVMN neurons in the intermediate and caudal regions of the nucleus. Arrows point at the efferent DVMN fibres. XII, hypoglossal motor nucleus. eGFP-IR, eGFP immunoreactivity. CC, central canal; **B**, Averaged distribution of transduced DVMN neurons expressing eGFP (left panel) and AlstR/eGFP (right panel) 5 weeks after microinjections of LVV-PRsX8-AlstR-IRES-eGFP (n=6) or LVV-PRsX8-eGFP (n=6). Diagrams illustrate the average numbers of neurons identified to express the respective transgene in one 30 μm slice taken from the rostral (<13.3 mm caudal from Bregma), intermediate (13.3-14.0 mm caudal from Bregma) and caudal (>14.0 mm caudal from Bregma) regions of the DVMN. Each symbol represents three transduced cells. 1633±100 eGFP- and 1074±79 AlstR/eGFP-expressing neurons were identified along the rostro-caudal extent of the left and right DVMN. No significant specific eGFP-labelling was observed outside the DVMN. 4V, fourth ventricle. AP, area postrema. NTS, nucleus of the solitary tract. From *J Physiol* 594: 4017, 2016.

Figure 4

Photomicrographs of coronal sections of the rat brainstem taken at low (left) and high (right) magnification illustrating representative examples of ChIEFtdTomato expression in the caudal region of the DVMN (Bregma level: -13.8 mm) 6 weeks after microinjections of PRsX8-ChIEFtdTomato-LVV. Neurons display specific membrane localization of the transgene. Arrows point at ventrally projecting axons of the transduced neurons (forming the vagus nerve). Images taken from this submission.

Critique: What percentage of the DVMN neurons that were targeted and/or studied project to the heart? The vast majority of DVMN neurons project to visceral targets within the respiratory system and gastrointestinal track, in addition to the minority that project to the heart. Since the experiments were focused upon cardiac function an assessment of the DVMN neurons that project to the heart is likely to be required.

Response: We respectfully disagree with the reviewer here. Earlier studies using neuronal traces identified significant numbers of the DVMN neurons labelled following pseudorabies virus injections into the cardiac ventricles or the apex of the heart (see summary Figure 7 in *J Neurosci* 15:1998, 1995). Our functional study published earlier this year demonstrated that the vagal preganglionic neurons which modulate ventricular contractility are located in the caudal left DVMN (*J Physiol* 594: 4017, 2016). Therefore, in this study we targeted the caudal aspect of the DVMN bilaterally, as illustrated by Figure 1b and Figure 2a. We believe that further assessment of DVMN neurons which specifically project to the heart is beyond the scope of this study aimed to investigate the causality between vagal tone and exercise capacity.

Critique: Results shown in Figs 1e and supplemental Fig 1 require additional explanation. Upon studying these figures, it appears that DVMN silencing provides almost complete beta-adrenergic activation before the addition of dobutamine (Supp Fig 1). LVESP is approx 165mmHg at paced baseline with allatostatin and is almost the same without allatostatin but with dobutamine. A similar result was observed in the contractility measurements (Fig 1e). LV ejection fraction increased with DVMN activation, presumably due to increased relaxation and increased EDV? How did contractility change during DVMN activation?

Response: We agree with the reviewer here. In our earlier study (*J Physiol* 594: 4017, 2016) we reported that acute inhibition of the DVMN neurons expressing AlstR following administration of allatostatin results in a moderate, but significant increase in LV contractility. At the same time, in conditions of DVMN inhibition, no further increase in LV contractility was observed in response to β -adrenoceptor stimulation with dobutamine (Figure 1e). For this resubmission we performed additional experiments which suggested that in addition to the small direct negative inotropic effect, DVMN neuronal projections strongly modulate the efficacy of β -adrenoceptor signalling and, therefore, sympathetic influences by controlling the level of GRK2 and β -Arrestin 2 expression by ventricular myocytes. In the revised manuscript we now report the new data and provide additional discussion. Please also see our detailed response to the next comment.

Critique: It would be informative for the authors to present a working mechanistic theory and explain the primary signaling pathways that motivate such powerful vagal modulation of cardiac reactivity to catecholamines, as stated in paragraph #6. Specific physiological mechanisms for the interaction between vagal tone and beta-receptor activation are not discussed and would enrich the manuscript.

Response: We fully agree with the reviewer and for this resubmission experimentally explored one of the potential mechanisms. Enhanced cardiac contractility (augmented inotropic state) following optogenetic recruitment of the DVMN neuronal projections suggested that the heart became more responsive to β -adrenoceptor stimulation. Therefore, we compared the level of expression of G protein-coupled receptor kinase 2 (GRK2) and β -Arrestin 2 in left ventricular myocytes of rats subjected to four daily sessions of vagal stimulation by optogenetic recruitment of the DVMN activity and animals which received sham stimulation. GRKs and arrestins are key negative regulators of GPCR (including β -adrenoceptor)-mediated signalling (*Mol Pharmacol*, 63, 9, 2003). GRKs promote phosphorylation of the intracellular domain of the active

receptor, recruiting arrestins to block GPCR coupling to G proteins resulting in receptor desensitization and internalization. In this revised submission we now include the new data (Figure 2f) which demonstrate that optogenetic stimulation of the DVMN activity markedly reduces the level of both GRK2 and β -Arrestin 2 expression in left ventricular myocytes. From these data we conclude that DVMN neuronal projections to the left ventricle control the expression of GRK2 and arrestins, and, therefore, modulate responsiveness of cardiomyocytes to β -adrenoceptor stimulation, control ventricular contractility and determine the exercise capacity. This conclusion is also supported by the results of our study published earlier this year which used a rat model of baroreflex dysfunction and demonstrated a clear association between parasympathetic dysfunction, impaired cardiac contractility and increased left ventricular GRK2 expression (Ackland et al. *Crit Care Med* 44: e614, 2016), although exercise capacity was not assessed. This notion is also fully consistent with the established role of GRK2 in the heart (*Mol Pharmacol*, 63, 9, 2003) and data on cross-talk between muscarinic and adrenoceptor-mediated signalling (see for example *Mol Pharmacol* 56, 813, 1999).

Other comments:

A. Originality and interest: the studies are novel and should be of interest to a wide audience.

Thank you!

B. Data & methodology: The primary limitation is the perceived non-specificity of the genetic targeting of the DVMN vagal pre-ganglionic neurons. This must be addressed.

Please see our detailed response above.

C. Appropriate use of statistics and treatment of uncertainties: Statistics were appropriate.

Thank you!

D. Conclusions: robustness, validity, reliability: Reliability could be tainted by the non-specificity limitation.

Please see our detailed response above.

E. Suggested improvements: Other suggestions provided above.

Please see our detailed response above.

F. References: References are appropriate.

Thank you!

G. Clarity and context: The text is clearly written and the figures nicely present the data.

Thank you!

Reviewers' comments:

Reviewer #1 (Remarks to the Author):

The authors have answered many of my questions, and while the idea that cardiac sympathetic activity correlates with enhanced exercise capacity is not novel, the paper has been significantly improved by the biochemical demonstration that DVMN activity resulted in a significant downregulation of GRK2 and beta-arrestin 2 expression in the LV. However, there remain two issues that have not been adequately addressed.

1. As I commented previously, the relationship between the DVMN firing frequency and exercise distance presented in Figure 1a is not linear but biphasic; there is a substantial range of frequencies (1.7 -2.7 Hz) at which the relationship is negative. In contrast, the other experiments suggest a somewhat linear relationship between vagal input and exercise capacity. The authors' response that "The shape of this relationship is similar to that of many curves one may find in every good Physiology textbook" is patronizing and certainly not acceptable. Also, their hypothesis that very high resting DVMN activity may limit cardiac contractility via direct actions of acetylcholine on ventricular myocytes" is also not acceptable as there is no evidence supporting that idea. In fact, I was unable to find any such evidence even in the Machhada et al. (J Physiol. 594: 4017, 2016) article the authors have cited.

Therefore, unless otherwise demonstrated, the biphasic curve relating DVMN firing frequency to the amount of voluntary exercise performed by the rats continues to raise important questions about how specifically and directly DVMN vagal preganglionic neurons control exercise capacity. First, it is important to remember that, in addition to vago-sympathetic interaction, indirect negative inotropic effects mediated by a positive force-frequency relationship are known to occur during vagal stimulation (Levy MN et al, Circ Res 1966; Circ Res 1976). In addition, species differences in parasympathetic innervation density and muscarinic receptor density in the ventricle seem to play an important role. In rodent hearts vagal innervation density is low and the direct effect of parasympathetic stimulation on LV contractility is known to be negligibly small. Thus, one would have to administer a substantial amount of acetylcholine directly to the ventricles to be able to observe a significant response. Second, it is now well established that an intrinsic ganglionated nerve plexus in the mammalian atria provides an integrative neuronal network, which modulates extrinsic autonomic projections to the heart and mediates the regulation of heart rate, atrioventricular nodal conduction and atrial and ventricular contraction (Rysevaite K, Heart Rhythm 2011). To what extent such a plexus intervenes as a determinant in the vagal control of exercise capacity has not been established. Altogether, the question of the biphasic response of exercise capacity to DVMN firing frequency remains unsettled; it appears more complex, and perhaps more important than claimed by the authors, and needs to be addressed.

2. Also, regarding the issue of the specificity, I commented previously that a more robust and direct way to demonstrate the effects would be to obtain DVMN recordings in brain slices after genetic manipulations (i.e., silencing and enhancement). I am puzzled by the authors response regarding the reversibility of the treatments and the improbability that the differences in resting DVMN activity would be observed following isolation of the DVMN in a slice preparation." As far as I know, lentiviral expression is used for long term expression. Since lentiviral constructs are being used they should be able to see changes in slices prepared from control and infected animals. Am I missing something?

Reviewer #2 (Remarks to the Author):

The authors have addressed each concern by providing additional data, and clarifying issues regarding the interpretation of their results and their novelty.

It is clear from their images that DVMNs are indeed infected with the lentivirus. While the possibility that adjacent neurons are also infected cannot be completely ruled out, there is a very high probability that it is the silencing of the DVMNs that is responsible for the phenotype of impaired exercise capacity. This is a valuable contribution to the fields of neuroscience and exercise physiology.

Reviewer #3 (Remarks to the Author):

The revised manuscript has improved considerably, with several aspects of the beta-adrenergic response explained more completely and additional insight regarding the mechanistic basis of GRK2 and beta-Arrestin 2 expression levels. Overall, the manuscript is very interesting, the figures are clearly presented, and the story is well-positioned.

One weakness remains. This reviewer reaffirms that the PRSx8 promoter is not sufficiently selective for the conclusions made in this study. Published literature not only shows this promoter drives expression in the retrotrapezoid nucleus (Eur J Neurosci. 2015 Sep;42(6):2271-82) and locus coeruleus (J Neurosci. 2015 Jan 28;35(4):1343-53) but most importantly for this work – also drives expression in the neighboring NTS (Cardiovasc Res. 2013 Nov 1;100(2):181-91). In this previously published work the PRSx8 promoter was also used to drive robust expression of AT1Rs in the NTS. This lack of selectivity of the PRSx8 promoter is a significant concern of this study.

It is surprising that the authors did not overcome this lack of selectivity with other approaches - such as injections into the cardiac ganglia with selective retrograde Cre-expressing viruses – such as PRV. They could combine this with a second floxed virus injected into the DMNX that would produce selective expression in DMNX neurons that project to the cardiac ganglia, rather than the current non-selective approach.

Additionally, the reviewer reaffirms that there is considerable evidence in the literature that the DMNX plays a critical role in the control of GI and respiratory function – to state that neurons in the DMNX are primarily parasympathetic neurons projecting to the heart is a significant oversimplification. If the authors performed the experiments this reviewer suggested, based upon this reviewer's review of the literature, they would likely find that only a small minority of DMNX neurons project to the cardiac ganglia, with a majority projecting to GI and respiratory targets. Although the cardiac parasympathetic neurons were certainly activated in their studies, which is the basis of the presented results, activating the entire DMNX is indeed a non-selective approach.

MS ID: NCOMMS-16-14931A-Z
Responses to referees' comments

Reviewer #1

The authors have answered many of my questions, and while the idea that cardiac sympathetic activity correlates with enhanced exercise capacity is not novel, the paper has been significantly improved by the biochemical demonstration that DVMN activity resulted in a significant downregulation of GRK2 and beta-arrestin 2 expression in the LV. However, there remain two issues that have not been adequately addressed.

Response: We thank this referee for his/her time taken to review our manuscript and overall positive assessment of our work. Below we provide our responses to the remaining criticisms and submit the second revision of our manuscript.

Critique: As I commented previously, the relationship between the DVMN firing frequency and exercise distance presented in Figure 1a is not linear but biphasic; there is a substantial range of frequencies (1.7 -2.7 Hz) at which the relationship is negative. In contrast, the other experiments suggest a somewhat linear relationship between vagal input and exercise capacity. The authors' response that "The shape of this relationship is similar to that of many curves one may find in every good Physiology textbook" is patronizing and certainly not acceptable. Also, their hypothesis that very high resting DVMN activity may limit cardiac contractility via direct actions of acetylcholine on ventricular myocytes" is also not acceptable as there is no evidence supporting that idea. In fact, I was unable to find any such evidence even in the Machhada et al. (*J Physiol.* 594: 4017, 2016) article the authors have cited.

Response: We thank the reviewer for this comment and sincerely apologize as our argument was not intended to appear patronizing. In the Machhada et al paper (*J Physiol.* 594: 4017, 2016) we reported that strong activation of a subset of the DVMN neurons (microinjections of glutamate) reduces left ventricular contractility in an anaesthetized (urethane) rat model.

Critique: Therefore, unless otherwise demonstrated, the biphasic curve relating DVMN firing frequency to the amount of voluntary exercise performed by the rats continues to raise important questions about how specifically and directly DVMN vagal preganglionic neurons control exercise capacity. First, it is important to remember that, in addition to vago-sympathetic interaction, indirect negative inotropic effects mediated by a positive force-frequency relationship are known to occur during vagal stimulation (Levy MN et al, *Circ Res* 1966; *Circ Res* 1976). In addition, species differences in parasympathetic innervation density and muscarinic receptor density in the ventricle seem to play an important role. In rodent hearts vagal innervation density is low and the direct effect of parasympathetic stimulation on LV contractility is known to be negligibly small. Thus, one would have to administer a substantial amount of acetylcholine directly to the ventricles to be able to observe a significant response.

Response: We thank the reviewer for this comment. We experimentally addressed this issue in our recent study conducted in anaesthetized rats (*J Physiol.* 594: 4017, 2016). We reported that tonic inhibitory muscarinic influence on cardiac inotropy is preserved in rats kept under urethane anesthesia and appears to be highly sensitive to the type of the anesthetic agent used (pentobarbitone abolishes inotropic vagal tone). We also reported that direct effect of DVMN activation on left ventricular contractility in rats is indeed small, but not insignificant (acute DVMN activation by glutamate microinjections

leads to ~10% reduction in LV contractility under the conditions of systemic sympathetic blockade).

Critique: Second, it is now well established that an intrinsic ganglionated nerve plexus in the mammalian atria provides an integrative neuronal network, which modulates extrinsic autonomic projections to the heart and mediates the regulation of heart rate, atrioventricular nodal conduction and atrial and ventricular contraction (Rysevaite K, Heart Rhythm 2011). To what extent such a plexus intervenes as a determinant in the vagal control of exercise capacity has not been established. Altogether, the question of the biphasic response of exercise capacity to DVMN firing frequency remains unsettled; it appears more complex, and perhaps more important than claimed by the authors, and needs to be addressed.

Response: We certainly agree with the reviewer here. Figure 1a of our original submission reported data obtained from the *in vitro* recordings of 115 DVMN neurons from 14 mice (8 cells on average). We start presentation of our material with the results of this *in vitro* experiment which was designed to address a simple question whether in the naïve animals an association exists between (global) vagal tone and voluntary exercise performance. As the reviewer correctly points out, the majority of the recorded neurons may not necessarily provide direct cardiac innervation, but at the group level mean DVMN activity is expected to reflect the general strength of the vagal tone. What our data suggest is that there is a relationship between the 'resting' intrinsic (there is no afferent input to these neurons in the isolated brainstem slices) vagal activity for a given individual and their voluntarily performed exercise. This is then also strongly corroborated by our interventional experiments using optogenetic stimulation or pharmacological inhibition of the DVMN activity. Indeed, it appears that if the mean resting DVMN discharge rate increases beyond ~2Hz the amount of voluntarily performed exercise decreases. It is currently unclear to us why this occurs, and we would like to emphasize that this is currently a simple association and establishing causality is beyond the scope of this focused experiment. Potential reasons are numerous (including these identified by the referee as well as various parameters under vagal control including intestinal function) and impossible to address experimentally given the time window when we can resubmit the next (and final) revision of our manuscript. To address this point of the reviewer we now include the following sentence in the revised version of the manuscript:

"Although, at present the reasons underlying this biphasic relationship remain unclear, lower amount of voluntary exercise associated with higher discharge rate of the DVMN neurons could be potentially explained by negative inotropic¹⁴ and chronotropic influences and/or non-cardiac effects of very high vagal activity".

We believe that this dataset is important and would be of interest to the readers of our article, however, we are prepared to remove it if the reviewer strongly disagrees with our reasoning.

Critique: Also, regarding the issue of the specificity, I commented previously that a more robust and direct way to demonstrate the effects would be to obtain DVMN recordings in brain slices after genetic manipulations (i.e., silencing and enhancement). I am puzzled by the authors response regarding the reversibility of the treatments and the improbability that the differences in resting DVMN activity would be observed following isolation of the DVMN in a slice preparation." As far as I know, lentiviral expression is used for long term expression. Since lentiviral constructs are being used they should be able to see changes in slices prepared from control and infected animals. Am I missing something?

Response: The reviewer is absolutely correct that lentiviral expression of transgenes is stable over long periods of time and fluorescently-labelled transduced cells can be identified and recorded in slice preparations. We believe that here the reviewer is asking to provide the confirmation that our experimental manipulations indeed alter the activity of DVMN neurons in the expected way. However, we have demonstrated exactly this in detail in our previous publications, and it seems futile to duplicate these data. The effect of acute administration of allatostatin acting on neurons transduced to express allatostatin receptor (AlstR) is readily reversible (please see the **Figure** below). Light pulses trigger precisely timed depolarizations and action potential firing of neurons expressing ChIEF and clear effects on cell physiology are only observed upon illumination. It is highly unlikely that acute allatostatin-induced silencing or 15 min-long 4 daily stimulations of the DVMN neurons *in vivo* would result in long-lasting significant changes in cell physiology to be detectable by standard electrophysiological recordings *in vitro* in the absence of these acute stimuli. In our previous publications we provided full validation of our approaches and reported data showing the effect of allatostatin on the DVMN neurons expressing AlstR and the effect of 445 nm light pulses on the DVMN neurons expressing light-sensitive channel ChIEF. Below we reproduce illustrations from these earlier publications:

(**A**) Representative whole-cell current-clamp recording from ChIEFtdTomato-expressing DVMN neuron illustrating depolarization and action potential firing in response to blue light (20 ms pulses). Four consecutive traces are overlaid. Action potential was elicited in response to 15 out of 16 pulses; (**B**) DVMN neurons expressing ChIEFtdTomato. Scale bar = 30 μm; (**C**) Mean data from voltage-clamp recordings (n=6) at a holding potential of -50 mV demonstrating the relationship between duration of the light stimulus and the normalized amplitude of inward current elicited by opening ChIEF channel; (**D**) Representative cell-attached recording from an AlstR/eGFP-positive DVMN neuron illustrating its reversible silencing in response to allatostatin (0.5 μM); (**E**) Current-voltage relationship (IV) of allatostatin-induced current. In this example, transduced DVMN neuron was voltage-clamped to -30 mV and hyperpolarizing voltage ramps to -130 mV (700 ms duration) were applied before and during allatostatin application. The displayed IV was obtained by subtracting the whole-cell IV obtained under control conditions, from that obtained in the presence of allatostatin (From *Cardiovasc Res* 95, 487, 2012).

Reviewer #2

The authors have addressed each concern by providing additional data, and clarifying issues regarding the interpretation of their results and their novelty. It is clear from their images that DVMNs are indeed infected with the lentivirus. While the possibility that adjacent neurons are also infected cannot be completely ruled out, there is a very high probability that it is the silencing of the DVMNs that is responsible for the phenotype of impaired exercise capacity. This is a valuable contribution to the fields of neuroscience and exercise physiology.

Response: We would like to thank this referee again for their extremely positive assessment of our work. We also thank this reviewer for acknowledging the specificity and validity of our experimental approach.

Reviewer #3

The revised manuscript has improved considerably, with several aspects of the beta-adrenergic response explained more completely and additional insight regarding the mechanistic basis of GRK2 and beta-Arrestin 2 expression levels. Overall, the manuscript is very interesting, the figures are clearly presented, and the story is well-positioned.

Response: We would like to thank this referee for his/her time taken to evaluate our revised submission and overall positive assessment of our work.

Critique: One weakness remains. This reviewer reaffirms that the PRSx8 promoter is not sufficiently selective for the conclusions made in this study. Published literature not only shows this promoter drives expression in the retrotrapezoid nucleus (Eur J Neurosci. 2015 Sep;42(6):2271-82) and locus coeruleus (J Neurosci. 2015 Jan 28;35(4):1343-53) but most importantly for this work – also drives expression in the neighboring NTS (Cardiovasc Res. 2013 Nov 1;100(2):181-91). In this previously published work the PRSx8 promoter was also used to drive robust expression of AT1Rs in the NTS. This lack of selectivity of the PRSx8 promoter is a significant concern of this study. It is surprising that the authors did not overcome this lack of selectivity with other approaches – such as injections into the cardiac ganglia with selective retrograde Cre-expressing viruses – such as PRV. They could combine this with a second floxed virus injected into the DMNX that would produce selective expression in DMNX neurons that project to the cardiac ganglia, rather than the current non-selective approach.

Response: We completely agree with this reviewer that the PRSx8 promoter is not exclusively selective for the DVMN neurons. However, one needs to take into account that we are not using transgenic mice where the selectivity of expression solely depends on the introduced transgene and the site of integration. Lentiviruses we use do not travel retrogradely (discussed in detail in our previous publications [Lonergan et al., 2005]) and even though PRSx8 promoter can be active in locus coeruleus and other brainstem neurons, such as RTN/C1 cells of the ventral medulla, carefully placed microinjections into the DVMN completely negate this factor, since the virus simply does not get to these remote areas.

Therefore, we respectfully disagree that this promoter is not sufficiently selective for DVMN targeting. We are confident that the required level of selectivity is effectively achieved by precise stereotaxic delivery of the viral vectors below the ventral anatomical

border of the DVMN and high affinity of the vector in transducing the DVMN neurons. Histological examination of the expression profile clearly shows that the transgenes are expressed in the targeted neuronal population. As we mentioned in our previous rebuttal letter, our group was the first to use PRSx8 promoter to target autonomic brainstem neurons and we have an extensive experience with it. Please find below the list of our key publications where we describe the results of several studies which used viral vectors with PRSx8 promoter to target distinct groups of the brainstem neurons. We targeted all the areas mentioned by the reviewer (RTN, LC, NTS) as well as C1 catecholaminergic group and the DVMN.

As we also mentioned and discussed previously, not all cells which express Phox2 efficiently drive PRSx8 promoter, hence the presence of this transcriptional factor as such is not yet a guarantee for the appearance of the designed transgenes. In the dorsal brainstem only DVMN and A2 catecholaminergic neurons of the neighbouring NTS express Phox2 transcription factors, therefore PRSx8 promoter can only be active in these two populations of cells. A2 neurons can in principle express transgenes driven by PRSx8 promoter and we attempted to use RPSx8 *in vivo* to target A2 neurons in our earlier studies (*Cardiovasc Res* 76: 184, 2007) by injecting adenoviral vectors **directly** into the NTS. Already in these early experiments we documented that the DVMN neurons are much more sensitive to PRSx8-bearing vectors. When PRSx8-driven constructs are placed in the lentiviral vectors, as in the present work, and injections are made into the DVMN or immediately ventral to its anatomical boundary, expression is essentially completely selective to the DVMN. This is easy to see since A2 catecholaminergic neurons project rostrally and do not overlap with the DVMN population.

We now kindly ask this reviewer to evaluate the expression profile of transgenes in the DVMN achieved following microinjections of viral vectors with PRSx8 promoter in our studies:

From: *Cardiovasc Res* **95**: 487, 2012

Panel A demonstrates that the vast majority (if not all) of dorsal brainstem neurons transduced to express AlstR/eGFP express choline acetyltransferase (ChAT), i.e. these cells are cholinergic neurons. In this area of the brainstem only DVMN and hypoglossal motoneurons are cholinergic. Hypoglossal neurons (XII nucleus) were indeed identified by ChAT immunoreactivity, but were not transduced as they do not express Phox2. **Panel B** shows an example of distribution of AlstR/eGFP-transduced DVMN neurons in relation to the location of A2 noradrenergic cells identified by tyrosine hydroxylase (TH) immunohistochemistry. Along with strong expression of AlstR/eGFP in the DVMN only occasional noradrenergic neurons were found to be transduced.

From: *J Physiol* **594**: 4017, 2016

Photomicrographs of the coronal sections of the rat brainstem showing expression of eGFP in an animal injected with LVV-PRsX8-eGFP. Images illustrate representative example of the distribution of transduced DVMN neurons in the intermediate and caudal regions of the nucleus. Arrows point at the efferent DVMN fibres. *Right*: Distribution of transduced DVMN neurons expressing eGFP and AlstR/eGFP five weeks after microinjections of LVV-PRsX8-AlstR-IRES-eGFP (n=6) or LVV-PRsX8-eGFP (n=6). Diagrams illustrate the average numbers of neurons identified to express the respective transgene in one 30 μm slice taken from the rostral, intermediate, and caudal regions of the DVMN. Each symbol represents three transduced cells. 1633 ± 100 eGFP- and 1074 ± 79 AlstR/eGFP-expressing neurons were identified along the rostro-caudal extent of the left and right DVMN. No significant specific eGFP-labelling was observed outside the DVMN. 4V, fourth ventricle. AP, area postrema. NTS, nucleus of the solitary tract.

This submission:

Photomicrographs of coronal sections of the rat brainstem taken at low (left) and high (right) magnification illustrating representative examples of ChIEFtdTomato expression in the caudal region of the DVMN (Bregma level: -13.8 mm) 6 weeks after microinjections of PRsX8-ChIEFtdTomato-LVV. Neurons display specific membrane localization of the transgene. Arrows point at ventrally projecting efferent fibers of the transduced neurons (forming the vagus nerve).

Now we kindly invite this reviewer to compare these images to representative (and only) example of transgene expression in the “NTS” taken from the study quoted by the referee in support of his/her criticism that PRSx8-based viruses are not specific and transduce NTS neurons (*Cardiovasc Res.* 100: 181, 2013):

Figure legend: Photomicrographs of coronal section of the NTS from an AT 1A^{-/-} mouse microinjected with green fluorescent protein virus (GFPv) showing immunofluorescent localization of tyrosine hydroxylase (left, TH, red), green fluorescent protein (middle, GFP, green), and the merged image (right, green and red). The GFP expression occurs in response to microinjection of the lentivirus with transgene expression under the control of the PRSx8 promoter. From: *Cardiovasc Res.* 100: 181, 2013.

Text description: Microinjections of GFPv resulted in strong expression of GFP in ventral regions of the NTS. Detailed counts throughout the NTS showed that 22±4% of TH-immunoreactive cells expressed GFP and 13±2% of GFP expressing cells were TH-immunoreactive (Figure 1B).

If PRSx8 promoter is active in the NTS neurons, then why in that study despite 4 microinjections of the virus **directly targeting** the NTS only the “ventral regions” (these which are adjacent(!) to the DVMN) appear to be transduced? Second, in their own description, the authors of that study indicate that only 13±2% of GFP expressing cells were found to be TH-immunoreactive, which implies that the remaining 87% of transduced neurons were in fact the DVMN neurons (since in this area of the brainstem only A2 and DVMN neurons express Phox2 and can be in principle transduced with PRSx8-bearing viruses). Careful examination of these published images (with all due respect co-localization of fluorescence on the provided images is not convincing) confirmed our own earlier observations that PRSx8 promoter is very weak to drive transgene expression in the NTS and despite direct targeting of the NTS the vast majority (if not all) of the transduced neurons are the DVMN neurons. In our study we avoid possible NTS transfection by placing microinjections below the ventral anatomical border of the DVMN, as emphasized in the revised version of our manuscript.

We also respectfully disagree with this reviewer that the experiment with injections of selective retrograde Cre-expressing viruses into the cardiac ganglia will provide higher level of selectivity in targeting vagal efferent projections. This approach is not trivial even if theoretically feasible. Delivery of retrograde Cre-expressing viruses into the cardiac ganglia would also require microinjections of the second virus with a CRE-dependent cassette into the DVMN. It would be very difficult to achieve robust expression of the optogenetic actuator and at the same time avoid cellular damage by CRE (over)expression. As such, retrograde viral transduction of cardiac ganglia is not a very well established protocol.

Critique: Additionally, the reviewer reaffirms that there is considerable evidence in the literature that the DMNX plays a critical role in the control of GI and respiratory function – to state that neurons in the DMNX are primarily parasympathetic neurons projecting to the heart is a significant oversimplification. If the authors performed the experiments this reviewer suggested, based upon this reviewer’s review of the literature, they would likely find that only a small minority of DMNX neurons project to the cardiac ganglia, with a majority projecting to GI and respiratory targets. Although the cardiac parasympathetic neurons were certainly activated in their studies, which is the basis of the presented results, activating the entire DMNX is indeed a non-selective approach.

Response: We agree and acknowledge that the DVMN neuronal projections also target the respiratory system and various visceral organs. Our functional study published earlier this year demonstrated that the vagal preganglionic neurons which modulate ventricular contractility are located in the caudal regions of the DVMN (*J Physiol* 594: 4017, 2016). Therefore, in this study to recruit these cardiac projections we targeted the caudal aspects of the DVMN (as illustrated by Figure 2a). The data obtained suggest that DVMN inhibition or activation result in functional and transcriptional changes at the level of the myocardium and these changes are associated with altered exercise capacity. Although we focused on the heart and see a clear cardiac phenotype, the main aim of this study was to experimentally investigate the causality between global parasympathetic vagal tone and exercise capacity. Therefore, we strongly believe that our experimental design is providing the required level of selectivity and entirely appropriate to address this question (even though some of the observed changes in cardiac physiology might be due to the recruitment of vagal efferent fibers projecting to other targets). We agree with the reviewer and in order to address this comment include the following text in the second revision of our manuscript:

"We hypothesize that high parasympathetic vagal tone generated by the DVMN neurons maintains the ability of the heart to mount an augmented contractile response to sympathetic stimulation and increases the 'operational range' of the heart by downregulating GRK2 and arrestin expression in ventricular myocytes. This tonic vagal influence originating from the DVMN appears to be independent of relatively modest direct acetylcholine-mediated negative inotropic effect¹⁴ and chronotropic control of the heart, which is provided by another notable group of vagal preganglionic neurons residing in the nucleus ambiguus⁹. Vagal preganglionic neurons which innervate the left cardiac ventricle are located in the caudal region of the left DVMN¹⁴. Since DVMN neurons provide parasympathetic innervation of the respiratory system and various visceral organs, in order to preferentially recruit DVMN cardiac projections we targeted the caudal aspects of the nucleus. Although, DVMN inhibition or activation resulted in functional and transcriptional changes at the level of the myocardium, it is plausible that some of the observed changes in cardiac physiology might be due to the inhibition/recruitment of vagal efferent projections to other targets²³ and recruitment of circulating cardiotropic factor(s)²⁴."

LIST OF OUR STUDIES WHERE VIRAL VECTORS WITH THE PRSx8 PROMOTER WERE USED:

1. Lonergan, T, Teschemacher, AG, Paton, JFR & Kasparov, S. (2004). Expression profile of adenoviral vectors incorporating hCMV, synapsin-1 and PRSx8 promoters in brainstem centres of cardiovascular control. *J Physiol*
<http://www.physoc.org/publications/proceedings/archive/index.asp>. An abstract where we described the first ever cellular profile of PRSx8 promoter expression in the brainstem in vivo.
2. Lonergan, T, Teschemacher, AG, Hwang, DY, Kim, KS, Pickering, AE & Kasparov, S. (2005). Targeting brain stem centers of cardiovascular control using adenoviral vectors: impact of promoters on transgene expression. *Physiol Genomics*, **20**, 165-172. First paper where we characterised the use of PRSx8 containing adenoviral vectors in the brainstem including retrograde expression from the spinal cord. This paper precipitated widespread interest in using PRSx8-based viral vectors by other research groups, including these of P. Guyenet (Virginia) and A.M. Allen (Melbourne, Australia).
3. Teschemacher, AG, Wang, S, Lonergan, T, Duale, H, Waki, H, Paton, JF & Kasparov, S. (2005). Targeting specific neuronal populations using adeno- and lentiviral vectors: applications for imaging and studies of cell function. *Exp Physiol*, **90**, 61-69. A review article where we also present additional data on the use of PRSx8-driven adenoviruses.
4. Teschemacher, AG, Paton, JF & Kasparov, S. (2005). Imaging living central neurones using viral gene transfer. *Adv Drug Deliv Rev*, **57**, 79-93. A review article where we discuss the use of PRSx8-driven viruses for imaging applications.
5. Duale, H, Waki, H, Howorth, P, Kasparov, S, Teschemacher, AG & Paton, JF. (2007). Restraining influence of A2 neurons in chronic control of arterial pressure in spontaneously hypertensive rats. *Cardiovasc Res*, **76**, 184-193. Our study where we used PRSx8-containing lentivirus to express a specific K⁺ channel in A2 neurons of the NTS. In that study we faced great difficulties trying to avoid expression in the DVMN and had to deliver injections at the very upper edge of the NTS. We had to discard a large number of animals because of the strong expression in the DVMN which was impossible to avoid. After that study we came to the conclusion that the use of PRSx8-based vectors for targeting NTS neurons in vivo is not appropriate.
6. Teschemacher, AG, Wang, S, Raizada, MK, Paton, JFR & Kasparov, S. (2008). Area-specific differences in transmitter release in central catecholaminergic neurons of spontaneously hypertensive rats. *Hypertension*, **52**, 1-8. In vitro application of PRSx8 promoter-based virus to fluorescently label NE-releasing neurons.
7. Kasparov, S & Teschemacher, AG. (2009). The use of viral gene transfer in studies of brainstem noradrenergic and serotonergic neurons. *Philos Trans R Soc Lond B Biol Sci*, **364**, 2565-2576. A review article where cell-specific targeting using PRSx8-based viruses is discussed in detail.
8. Gourine, AV, Kasymov, V, Marina, N, Tang, F, Figueiredo, MF, Lane, S, Teschemacher, AG, Spyer, KM, Deisseroth, K & Kasparov, S. (2010). Astrocytes control breathing through pH-dependent release of ATP. *Science*, **329**, 571-575. In this study an adenoviral vector with PRSx8 promoter was used to fluorescently label RTN neurons.
9. Marina N, Abdala AP, Trapp S, Li A, Nattie EE, Hewinson J, Smith JC, Paton JF, Gourine AV. (2010) Essential role of Phox2b-expressing ventrolateral brainstem neurons in the chemosensory control of inspiration and expiration. *J Neurosci*. **30**: 12466-12473. In this study a lentivirus with PRSx8 promoter was used to target the RTN neurons.
10. Marina N, Abdala AP, Korsak A, Simms AE, Allen AM, Paton JF, Gourine AV. (2011). Control of sympathetic vasomotor tone by catecholaminergic C1 neurones of the rostral ventrolateral medulla oblongata. *Cardiovasc Res* **91**: 703-710. In this study a lentivirus with PRSx8 promoter was used to target catecholaminergic C1 neurons.

11. Mastitskaya S, Marina N, Gourine A, Gilbey MP, Spyer KM, Teschemacher AG, Kasparov S, Trapp S, Ackland GL, Gourine AV. (2012). Cardioprotection evoked by remote ischaemic preconditioning is critically dependent on the activity of vagal pre-ganglionic neurones. *Cardiovasc Res* **95**: 487-494. *In this study a lentivirus with PRSx8 promoter was used to target the DVMN neurons.*
12. Tang, F, Lane, S, Korsak, A, Paton, JF, Gourine, AV, Kasparov, S & Teschemacher, AG. (2014). Lactate-mediated glia-neuronal signalling in the mammalian brain. *Nature Communications*, **5**, 3284. *In this study an adenovirus with PRSx8 promoter was used to fluorescently label neurons of the locus coeruleus.*
13. Machhada A, Ang R, Ackland GL, Ninkina N, Buchman VL, Lythgoe MF, Trapp S, Tinker A, Marina N, Gourine AV. (2015) Control of ventricular excitability by neurons of the dorsal motor nucleus of the vagus nerve. *Heart Rhythm* **12**: 2285-2293. *In this study a lentivirus with PRSx8 promoter was used to target the DVMN neurons.*
14. Machhada A, Marina N, Korsak A, Stuckey DJ, Lythgoe MF, Gourine AV. (2016). Origins of the vagal drive controlling left ventricular contractility. *J Physiol* **594**: 4017-4030. *In this study a lentiviral vector with PRSx8 promoter was used to target the DVMN neurons.*

REVIEWERS' COMMENTS:

Reviewer #1 (Remarks to the Author):

I am thankful to the authors for their attempt to answer my comments on figure 1a, which shows a non-linear (biphasic) relationship between the in vitro recordings of 115 DVMN neurons and the amount of voluntary exercise exhibited by the animals in a 24 h period, while in sharp contrast, the other experiments suggest a somewhat linear relationship between vagal input and exercise capacity. Unfortunately, their attempt to provide a rationale to support such results is unconvincing. In my opinion the figure remains highly problematic because it disagrees with the main conclusion of the paper that the data that parasympathetic vagal drive generated by the DVMN neurons determines the ability to exercise. It may do so in part, but only up to a point, and perhaps not specifically. Therefore, the figure should be deleted and the overall conclusion of the manuscript should be significantly toned down.

Reviewer #3 (Remarks to the Author):

Thank you for the detailed and comprehensive response. Congratulations on a valuable study!

MS ID: NCOMMS-16-14931B
Responses to referees' comments

Reviewer #1

I am thankful to the authors for their attempt to answer my comments on figure 1a, which shows a non-linear (biphasic) relationship between the in vitro recordings of 115 DVMN neurons and the amount of voluntary exercise exhibited by the animals in a 24 h period, while in sharp contrast, the other experiments suggest a somewhat linear relationship between vagal input and exercise capacity. Unfortunately, their attempt to provide a rationale to support such results is unconvincing. In my opinion the figure remains highly problematic because it disagrees with the main conclusion of the paper that the data that parasympathetic vagal drive generated by the DVMN neurons determines the ability to exercise. It may do so in part, but only up to a point, and perhaps not specifically. Therefore, the figure should be deleted and the overall conclusion of the manuscript should be significantly toned down.

Note from the Editor: "...You will see that referee #1 still has issues with the biphasic relationship between the in vitro recordings of DVMN neurons and the amount of voluntary exercise exhibited by the animals in a 24 h period, and she/he suggests this experimental result to be taken out and the manuscript to be toned down. While we agree that some toning down might be necessary, we still think that this result (presented in fig 1a) should remain part of the manuscript and that it should be carefully discussed".

Response: We would like to thank this referee for his/her time taken to evaluate our revised submission and overall positive assessment of our work. We also thank the Editors who share our view that this dataset is important and would be of interest to the readers of this article. As we argued in our previous response letter we start presentation of our material with the results of this *in vitro* experiment which was designed to address a simple question whether in the naïve animals an association exists between (global) vagal tone and voluntary exercise performance. As the reviewer correctly pointed out, the majority of the recorded neurons may not necessarily provide direct cardiac innervation, but at the group level mean DVMN activity is expected to reflect the general strength of the vagal tone. What our data suggest is that there is a relationship between the 'resting' intrinsic (there is no afferent input to these neurons in the isolated brainstem slices) vagal activity for a given individual and their voluntarily performed exercise. The reasons underlying this apparently biphasic relationship are currently unclear and we would like to emphasize that this is a simple association and establishing causality is beyond the scope of this focused experiment. To address this point of the reviewer we now include the following discussion in the revised version of the manuscript:

"This largely linear relationship did not continue with spontaneous mean firing rates of DVMN neurons above 2 Hz (recorded in brainstem slices of two animals) (Figure 1a). Although, at present the reasons underlying this biphasic relationship remain unclear, the lower amount of voluntary exercise associated with a higher discharge rate of the DVMN neurons could be potentially explained by negative inotropic¹⁴ and chronotropic influences as well as non-cardiac effects of high vagal activity. Considering that the majority of the vagal projections originating from the DVMN innervate visceral targets, it is conceivable that higher discharge rate of the DVMN neurons mimics the postprandial state and, thus, may reduce the motivation to exercise. As voluntary exercise is dependent on motivation, and motivation is determined by a multitude of factors, the design of the subsequent studies employed forced exercise experimental paradigm."

Reviewer #3

Thank you for the detailed and comprehensive response. Congratulations on a valuable study!

Response: We would like to thank this referee for his/her time taken to evaluate our revised submission and very positive assessment of our work.